# Aerosol type retrieval and uncertainty quantification from OMI data

Anu Kauppi<sup>1,2</sup>, Pekka Kolmonen<sup>1</sup>, Marko Laine<sup>1</sup>, and Johanna Tamminen<sup>1</sup>

<sup>1</sup>Finnish Meteorological Institute, Helsinki, Finland

<sup>2</sup>Department of Mathematics and Statistics, University of Helsinki, Helsinki, Finland

Correspondence to: Anu Kauppi (anu.kauppi@fmi.fi)

# Abstract.

We discuss uncertainty quantification for aerosol type selection in satellite-based atmospheric aerosol retrieval. The retrieval procedure uses pre-calculated aerosol microphysical models stored in look-up tables (LUTs) and top of atmosphere spectral reflectance measurements to solve the aerosol characteristics. The forward model approximations cause systematic differences

5 between the modeled and observed reflectance. Acknowledging this model discrepancy as a source of uncertainty allows us to produce more realistic uncertainty estimates and assists the selection of the most appropriate LUTs for each individual retrieval.

This paper focuses on the aerosol microphysical model selection and characterization of uncertainty in the retrieved aerosol type and aerosol optical depth (AOD). The concept of model evidence is used as a tool for model comparison. The method is based on Bayesian inference approach where all uncertainties are described as a posterior probability distribution. When there

- 10 is no single best matching aerosol microphysical model we use a statistical technique based on Bayesian model averaging to combine AOD posterior probability densities of the best fitting models to obtain an averaged AOD estimate. We also determine the shared evidence of the best matching models of a certain main aerosol type in order to quantify how plausible each main aerosol type is in representing the underlying atmospheric aerosol conditions.
- The developed method is applied to Ozone Monitoring Instrument (OMI) measurements using multi-wavelength approach for retrieving the aerosol type and AOD estimate with uncertainty quantification for cloud-free over-land pixels. Several larger pixel set areas were studied in order to investigate robustness of the developed method. We evaluated the retrieved AOD by comparison with ground-based measurements at example sites. We found that the uncertainty of AOD expressed by posterior probability distribution reflects the difficulty in model selection. The posterior probability distribution can provide a comprehensive characterization of the uncertainty in this kind of problem for aerosol type selection. As a result, the proposed method can account for the model error and also include the model selection uncertainty in the total uncertainty budget.

### 1 Introduction

The atmospheric aerosols play an important role in our understanding of Earth's climate system. Aerosols have direct and indirect influence on Earth's radiation budget. Satellite remote sensing observations have been utilized for years to provide information about atmospheric aerosol conditions in global scale. The spaceborn data are very useful for detecting and following

25 dynamic natural or anthropogenic events such as sand storms and active fires. The most common retrieved aerosol characteristic is the aerosol optical depth (AOD) which is a function of the loading, size distribution and optical properties of aerosol particles. There are a number of satellite instruments delivering aerosol products and providing aerosol characteristics e.g. the Ozone Monitoring Instrument (OMI) (Torres et al., 2007), the Moderate Resolution Imaging Spectroradiometer (MODIS) (Levy et al., 2010), the Global Ozone Monitoring Experiment-2 (GOME-2) (Hassinen et al., 2015), the Multi-angle Imaging SpectroRadiometer (MISR) (Kahn et al., 2010), the (Advanced) Along-Track Scanning Radiometers ((A)ATSR) (Thomas et

al., 2009; Kolmonen et al., 2016), the Cloud-Aerosol Lidar and Infrared Path finder (CALIPSO) (Winker et al., 2009), the Scanning Imaging Absorption spectroMeter for Atmospheric Chartography (SCIAMACHY) (Bovensmann et al., 1999), the Polarisation and Directionality of the Earth's Reflectance (POLDER) (Dubovik et al., 2011) and the Spinning Enhanced Visible and Infrared Imager (SEVIRI) (Govaerts et al., 2010; Wagner et al., 2010).

There is an increasing potential for using and incorporating satellite-based aerosol information as the instruments are getting

- better in resolution and more sophisticated for detecting aerosols (Holzer-Popp et al., 2013). In addition, the improvement of retrieval algorithms and the development of novel methodologies extend opportunities to use the data. Especially, one target is to derive information about small aerosol particles (diameter less than 1  $\mu$ m) from satellite measurements. An important and challenging use of satellite measurements is to assimilate aerosol characteristics into large-scale global aerosol models (Benedetti et al., 2009). Furthermore, the satellite based data can be combined with numerical models when estimating aerosol
- emission fluxes (Huneeus et al., 2012) or spatially constraining amount of aerosol emissions (Wang et al., 2012; Xu et al., 2013). Data validness as well as identification and quantification of uncertainties are acknowledged when data are used.

Uncertainties in satellite-based aerosol retrievals arise from many sources, e.g. cloud contamination, treatment of surface reflectance and instrumental issues. It is typical in the aerosol retrievals that the radiative transfer (i.e. forward model) calculations have been replaced by pre-calculated look-up tables (LUTs) in order to speed up the needed computations. The LUTs

- are often multi-dimensional tables containing simulated discrete descriptions of varying aerosol conditions. Aerosols can be classified into categories (i.e. main types) such as clean background, urban pollution, dust, smoke (from biomass burning) and sea salt based on the origins of the aerosol particles. The optical and microphysical properties of different aerosol types are described in corresponding LUTs. The aerosol properties in the LUTs can be based on observations or combination of observations and climate models (Holzer-Popp et al., 2013). The situation is more complicated for a retrieval algorithm when
- an aerosol containing air-mass is a mixture of different types, e.g. mixture of dust aerosols and biomass burning aerosols. The proper aerosol type selection from LUTs is a source of uncertainty and affects the accuracy of the retrieval. Povey et al. (2015) give an overview of the error analysis and representation of uncertainty in the satellite data. One application they discuss is related to the AOD retrievals where unquantifiable errors arise from the choice of a forward model (i.e. aerosol microphysical properties).
- In this paper we discuss characterization of uncertainty in the aerosol type and AOD retrieval. We utilize the method, described in Määttä et al. (2014), for estimating the uncertainty in the retrieved AOD due to the aerosol microphysical model selection and the approximations in forward modeling. The method is based on the Bayesian inference approach where uncertainty estimates are an inherent part of the formulation (MacKay, 1992; Spiegelhalter et al., 2002; Robert, 2007). The uncertainty is given as a posterior density function of the AOD and a point estimate for the AOD is the maximum a poste-
- 35 rior (MAP) value. We calculate the model evidence value for each aerosol microphysical model involved in order to compare

models and do the model selection. The selection of single best fitting aerosol microphysical model is not always clear and this uncertainty has also been addressed in this study. We calculate the averaged posterior probability distribution where the individual model posterior distributions are weighted by their evidence. This is implemented by the Bayesian model averaging technique (Hoeting et al., 1999). We also perform the shared evidence of the best matching models within main aerosol type

in order to quantify plausibility of each main aerosol type. We acknowledge the forward modeling uncertainty, i.e. model discrepancy (Kennedy et al., 2001; Brynjarsdóttir et al., 2014) which arises from non-modeled systematic differences between the modeled and observed reflectance. The described method is applied for the aerosol retrieval using cloud screened data from the OMI instrument.

The used data and methodology are introduced in Sect. 2 and 3. We have investigated the performance of the method with 10 case studies presented in Sect. 4. Section 5 discuss the features and possibilities of the method.

# 2 OMI data


The Dutch-Finnish OMI instrument is on board NASA's Earth Observing System (EOS) Aura platform which was launched in July 2004 (Levelt et al., 2006). The Aura satellite is in a polar sun-synchronous orbit crossing the equator approximately at 13:45 local time. OMI measures sunlight backscattered from the Earth in the ultraviolet (UV) and visible (VIS) wavelength bands (270-500 nm). The ground pixel size at nadir is  $13 \times 24 \text{ km}^2$ . The retrieved products include atmospheric trace gases (ozone, NO<sub>2</sub>, SO<sub>2</sub>, HCHO, BrO and OCIO), surface UV, cloud information and aerosol characteristic.

The two operational aerosol algorithms to retrieve aerosol characteristics from OMI measurements are the OMI near-UV aerosol data product (OMAERUV) and the OMI multi-wavelength aerosol data product (OMAERO) (Torres et al., 2007, 2002). OMAERUV uses in the retrieval two wavelength bands at 354 and 388 nm to determine the AOD, aerosol index and single

- 20 scattering albedo (Ahn et al., 2014). OMAERO uses the near UV and visible wavelengths between 330 and 500 nm providing the AOD, best matching aerosol model and aerosol characteristics associated with the best model (e.g. single scattering albedo and aerosol indices) (Curier et al., 2008). The retrievals of AOD and single scattering albedo from OMAERUV and OMAERO have been evaluated using air-borne sunphotometer, ground-based sun/sky radiometer and other satellite measurements (Ahn et al., 2008; Livingston et al., 2009).
- The OMI data used in this study have been extracted via Mirador data search tool provided by the NASA Goddard Earth Sciences Data and Information Services Center (GES DISC) data access system (https://urs.earthdata.nasa.gov). We calculated the top-of-the-atmosphere (TOA) spectral reflectance (referred to as measured or observed reflectance from now on) from the OMI Level 1B VIS and UV radiances and Level 1B Solar irradiance data. We took the effective cloud fraction information from the Level 2 OMI O<sub>2</sub>-O<sub>2</sub> cloud product (OMCLDO2). Then we applied a simple scheme by using 0.34 as an effective
- 30 cloud fraction threshold value for detecting and excluding a cloudy pixel. Thus we followed only one of three tests for cloud screening used by the OMAERO algorithm. The high threshold value of 0.34 was chosen in order to avoid excluding desert dust scenes (OMAERO Readme Document, 2011). To assure measurement data quality in the retrieval we used the pixel quality

and error flags from the OMI Level 1B radiance products. In addition, to ensure the forward model quality, we excluded data where solar zenith angle was above  $75^{\circ}$ .

We used GroundPixelQuality flag from the OMI Level 1B radiance product to choose the over land pixels as this study was concentrated on the aerosol types that are dominant over land areas. We accepted a pixel and specify it as land pixel if the

- flag indicated ground type to be land, shallow inland water, ocean coastline/lake shoreline, ephemeral (intermittent) water or deep inland water (OMI Level 1B Output products and Metadata, 2009). For surface reflectivity, we used the climatological surface reflectance database from the OMI Earth Surface Reflectance Climatology product OMLER (v003). The OMLER (v003) product data file (OMI-Aura\_L3-OMLER\_2005m01-2009m12\_v003-2010m0503t063707.he5) was extracted from the GES DISC data Service. The OMLER product contains in a 0.5 x 0.5 degree grid global maps of the monthly climatology of
- Lambert equivalent reflectance (LER) based on five years (2005-2009) of OMI data (OMLER Readme Document, 2010). In our analysis we have used about 1 nm wide wavelength bands centered at 342.5, 367.0, 376.5, 388.0, 399.5, 406.0, 416.0, 425.5, 436.5, 442.0, 451.5, 463.0 and 483.5 nm. These 13 bands include one wavelength in the UV region and the rest in the VIS region. The O2-O2 absorption wavelength band centered at 477 nm brings important information about the cloud height and for cloud-free scene about the aerosol layer height for high enough AOD levels (Veihelmann et al., 2007). However, we
- omitted in our study the band 477 nm due to experimental purpose and since we did not need aerosol height information.

the capability of the OMI multi-wavelength algorithm to distinguish between different aerosol types.

# 2.1 Aerosol microphysical models

The aerosol microphysical models stored in the OMI LUTs are produced via the radiative transfer calculations for a range of aerosol physical properties and sun-satellite geometries (Torres et al., 2002, 2007). There are four main aerosol types: weakly absorbing (WA), biomass burning (BB), desert dust (DD) and volcanic (VO) aerosols. The weakly absorbing type aerosol models are composed of urban-industrial and natural oceanic aerosols (Torres et al., 2002). Veihelmann et al. (2007) discuss

The main types are split into sub-types (i.e. models) according to the aerosol size distribution, refractive index and vertical profile. We used a set of OMI aerosol microphysical models, total of fifty models, in the work presented here (see Table 1). Each model consists of a set of parameters (e.g. AOD, single scattering albedo, viewing and solar zenith angle, relative azimuth angle, path reflectance, transmission and spherical albedo) with predefined values at node points.

A weakly absorbing aerosol model 'WA1114' represents sea salt particles having a higher fraction of coarse particles than the other weakly absorbing models (see Table 1). We have classified the model 'WA1114' as the fifth main aerosol type when reporting results from the case examples (see Sect. 4).

#### 3 Methodology



The proposed method is applied to the retrieval scheme that is similar to the OMAERO algorithm. The unknown aerosol parameter is the AOD at the reference wavelength of 500 nm, for which we will use symbol  $\tau$ . The related uncertainty is analyzed using Bayesian statistical inference. The observations are TOA reflectances  $\mathbf{R}_{obs}(\lambda)$  at a set of wavelengths  $\lambda =$ 

 $(\lambda_1, \dots, \lambda_n)$ . The modeled spectral reflectance  $\mathbf{R}_{mod}(\tau, \lambda)$  depends on  $\tau$  within the specific aerosol microphysical model in LUT. The AOD parameter  $\tau$  is adjusted between the nodal values in the model LUT to find the modeled reflectance that has the best fit with the observed spectral reflectance.

Assuming a Lambertian surface the contribution of the radiation at the TOA can be separated from that of the atmosphere 5 (e.g. Chandrasekhar (1960)) leading to the equation for modeled reflectance as

$$\boldsymbol{R}_{\text{mod}}\left(\lambda,\tau,\mu,\mu_{0},\Delta\phi,p_{s}\right) = R_{a}\left(\lambda,\tau,\mu,\mu_{0},\Delta\phi,p_{s}\right) + \frac{A_{s}(\lambda)}{1 - A_{s}(\lambda)s\left(\lambda,\tau,p_{s}\right)}T\left(\lambda,\tau,\mu,\mu_{0},p_{s}\right).$$
(1)

Here path reflectance  $R_a$ , transmittance T and spherical albedo s of the atmosphere are derived from LUT by interpolation as a function of  $\lambda$ ,  $\tau$ ,  $\Delta\phi$  (relative azimuth angle),  $p_s$  (surface pressure),  $\mu$  (cosine of viewing zenith angle) and  $\mu_0$  (cosine of solar zenith angle). The sun-satellite geometry data  $\Delta\phi$ ,  $p_s$ ,  $\mu$  and  $\mu_0$  are included in the OMI Level 1B data. The surface reflectance

10  $A_s$  is taken from the Lambertian equivalent surface reflectance climatology based on the geolocation of the retrieved pixel and month.

# 3.1 Acknowledging the model discrepancy

15

The aerosol microphysical models used in the retrieval procedure are discrete representations of the aerosols in the real atmosphere. Approximations in forward modeling together with uncertainties in the assumptions, e.g. in the surface reflectance, cause model discrepancy, which manifests itself as systematic deviations between the modeled and observed reflectance.

We pay special attention to the model discrepancy in the fitting process by adding the related uncertainty term  $\eta(\lambda)$  to the observation model

$$\boldsymbol{R}_{\text{obs}}(\lambda) = \boldsymbol{R}_{\text{mod}}(\tau, \lambda) + \eta(\lambda) + \epsilon_{\text{obs}}(\lambda).$$
<sup>(2)</sup>

The model discrepancy error term  $\eta(\lambda)$  enables correlated errors between neighboring wavelengths, thus allowing for smooth 20 departures from the model. The measurement error  $\epsilon_{obs}(\lambda) \sim N(0, \sigma_{obs}^2(\lambda))$  will describe the independent instrument noise that will be assumed to be known in the retrieval procedure coming from the instrument properties and from the calculation of the observed reflectance. In the fitting procedure, for simplicity, we have  $\sigma_{obs}(\lambda) = R_{obs}(\lambda)/SNR$ , where we used value SNR = 500 for the signal-to-noise ratio of the instrument.

Our approach to estimate the model discrepancy term η(λ) was to explore systematic differences between the measured and
modeled reflectance (i.e. residuals). The systematic structure in the residuals indicates inadequacy in the forward model. The
model discrepancy was characterized using a zero mean Gaussian process η(λ) ~ GP(0, C) (Rasmussen and Williams, 2006),
as described by Määttä et al. (2014), where the covariance matrix C defines the wavelength-dependent correlation properties of
the discrepancy. The covariance matrix C was constructed by means of an empirical semivariogram when the variances of the
residual differences were calculated for each wavelength pairs with the distance *d*. Next, the theoretical Gaussian variogram
model was fitted to these empirical semivariogram values. The outcome of this analysis were the values for parameters that
defines the model discrepancy covariance matrix C (see Määttä et al. (2014) for details).

We assume that the likelihood function describing the distribution of the observations given the model follows a Gaussian distribution. The likelihood function has an additional error covariance term due to the model error,

$$p(\boldsymbol{R}_{\text{obs}}|\tau,m) \propto \exp\left(-\frac{1}{2}\boldsymbol{R}_{\text{res}}(\lambda)^{T}\left(\mathbf{C} + \text{diag}\left(\sigma_{\text{obs}}^{2}(\lambda)\right)\right)^{-1}\boldsymbol{R}_{\text{res}}(\lambda)\right),\tag{3}$$

where *R*<sub>res</sub>(λ) = *R*<sub>obs</sub>(λ) - *R*<sub>mod</sub>(τ, λ) is the residual, C is the model discrepancy covariance matrix and diag(σ<sup>2</sup>(λ)) is a
diagonal matrix of the measurement error variances σ<sup>2</sup><sub>obs</sub>(λ). The likelihood function is needed for calculation of posterior distribution using Bayes' formula (see Sect. 3.2).

### 3.2 Aerosol type and AOD retrieval with uncertainty quantification


In the Bayesian inference, the solution of an inverse problem is presented as a posterior distribution of the unknown. This approach provides a natural way to present the uncertainty in the AOD and in the aerosol microphysical model m. By Bayes' formula the posterior distribution for  $\tau$  within the model m and given the observed reflectance  $\mathbf{R}_{obs}$  is

$$p(\tau | \boldsymbol{R}_{\text{obs}}, m) = \frac{p(\boldsymbol{R}_{\text{obs}} | \tau, m) p(\tau | m)}{p(\boldsymbol{R}_{\text{obs}} | m)},\tag{4}$$

where  $p(\mathbf{R}_{obs}|\tau, m)$  is the likelihood and  $p(\tau|m)$  is a prior distribution for  $\tau$  depending on the aerosol microphysical model m. The denominator  $p(\mathbf{R}_{obs}|m)$  does not depend on  $\tau$  and acts to normalize the numerator. We assumed that the prior  $p(\tau|m)$  follows a log-normal distribution in order to ensure that the estimated AOD is positive. The calculation of the actual posterior

distribution requires solving integrals over the parameter and model space. In our case, the model selection procedure seeks the solution for one parameter  $\tau$  and then the calculation of posterior distribution is fairly straightforward by numerical quadrature. The calculation of the posterior distribution is presented in more detail in Määttä et al. (2014).

The denominator  $p(\mathbf{R}_{obs}|m) = \int p(\mathbf{R}_{obs}|\tau, m) p(\tau|m) d\tau$  in Eq. (4) is the probability of the observed reflectance  $\mathbf{R}_{obs}$  assuming the model m is the correct one. However, when considering our problem of choosing the right model m, the  $p(\mathbf{R}_{obs}|m)$ 

acts as an evidence in favour for m. Consequently, we compare models using their evidence values. In the retrieval procedure we accept the models with the highest evidence until a cumulative sum of the selected models' evidences pass the value of 0.8 or the number of selected models is ten.

Since we assume that a priori all models are equally likely, we end up calculating the relative evidence for each selected model  $m_i$  by formula

$$p(m_i|\mathbf{R}_{obs}) = \frac{p(\mathbf{R}_{obs}|m_i)}{\sum_j p(\mathbf{R}_{obs}|m_j)}.$$
 (5)

Here the denominator is a sum over all the evidences of the models  $m_j$  under the comparison process (see Määttä et al. (2014) for details). The relative evidence indicates how plausible the aerosol microphysical model is among the set of potential models.

Even when a model has the highest evidence it does not ensure that it gives an adequate fit to the observed reflectance. The goodness of fit of the selected model is analyzed by the modified chi-squared value

$$\chi^2 = \frac{1}{n-1} \mathbf{R}_{\text{res}}(\lambda)^{\text{T}} \left( \mathbf{C} + \text{diag}(\sigma^2(\lambda)) \right)^{-1} \mathbf{R}_{\text{res}}(\lambda),$$
 (6)

where  $\mathbf{C}$  is a covariance matrix for the model discrepancy and n is the number of wavelength bands in the spectral reflectance. We accepted the retrieved solution (i.e. the selected best model) if this merit function gives a value