# Peer review of "Aerosol type retrieval and uncertainty quantification from OMI data"

_Atmospheric Measurement Techniques, 2017_

## Referee Comment (RC1) · Anonymous Referee #1 · 27 Mar 2017

The paper addresses a very important topic in aerosol retrievals from satellite: the uncertainty of the selected aerosol model and the impact on the AOD. Indeed, this is one of the largest uncertainties for aerosol AOD and the proposed method is very appropriate. It builds on current methods and elegantly generalises the techniques using a sound approach. The paper is well written and well structured and I feel this paper could give an important contribution to a more accurate retrieval of AOD from space based instruments.

There are a few improvement that I deem necessary for this paper to be acceptable. In its present form it lacks a clear definite conclusion and recommendation. A very decent physical and mathematical framework is presented, however at the end the reader is left with a somewhat unsatisfactory feeling, not knowing whether the whole exercise was successful or not. For me the questions that are addressed here are: 1) does the

[Figure]

AOD retrieval improve when a combination of aerosol models is allowed and combined using the Bayesian model evidence? 2) Does the model selection uncertainty give a better estimate of the AOD uncertainty than the current one?

The authors pose the questions and address them, but I see no clear answer for these questions. It's left hanging in the conclusion section. It says 'the posterior probability distribution can characterise the uncertainty more extensively than commonly given standard deviation'. Fair enough, but what does this mean? Is it better? Should we generally apply this method? Also from the provided sensitivity studies it is just not clear whether things work as expected (probably leading to the general inconclusive conclusion section).

What I lack is an answer to these questions (supported by evidence): Does the average AOD perform better than the standard one, when compared to AERONET? If not, is this reflected in a larger uncertainty? If yes, are the AERONET and OMI AOD retrievals consistent within this new the uncertainty?

If this could be adequately answered, i recommend this paper for publication.

Minor comments:

p1l24 (and a few more): data is -> data are

p3l25: referred -> referred to

p3l28: a cloudy ground pixel sounds strange. I would say a cloudy atmosphere pixel. Or just a cloudy pixel.

p3l31: What is a wise quality?

p4l4: Before the start of the new sentence, add 'For surface reflectivity,' (we used.. etc)

p4l11-13: you say: the band at 477 nm adds important info, yet you exclude it specifically. Why?

p5l3: Equation (1) is not just a 'formula'. Start this discussion with a physical description like: Assuming a Lambertian surface the contribution of the radiation at the TOA can be separated from that of the atmosphere (e,g. Chandrasekhar, 1960), viz. etc.

p5l10: of the real -> of the aerosols in the real

p5l10-11: This forward model app error,.. Which one? You haven't described an error yet. Do you mean the difference between real and approx. reflectances? Then describe that.

p5l13: This is strange: I would expect that a total (megs) error would be forward model error, noise (and perhaps more). Noise surely doesn't include forward model error? What is epsilon_obs? Noise or total? Rephrase l11.

p6. Increase the size of eq. 4 and 5, like eq. 1. They are the basis of the paper.

p7l19: cover -> covers

p7l20: cover -> covers a

p7l22-25: Move this to section 2. And add a description of MODIS, which is introduced in the next paragraph.

p8l1 & Figure 1. The OMI pixels -> The OMI pixels that were analysed The location of the OMI pixels within the MODIS swath are not clear. In Figure 1 add the contours of the OMI pixels that are used in Fig 2-6.

p8l2: The pixel has no data if -> No data are reported if the pixel is

p8l18: pixel wise -> pixel-wise

p823: in the latter day case -> On the 27th,

p9l8-10: Figure 8 is superfluous. Remove it and on describe the results from it in the text. It will reduce the number of figures, which is needed anyway.

p9l11-13;: Elaborate on this result. It is as important as the 16th.

p9l14-22: Here's the first missing conclusion. So you compared the Angstrom exponents. Whats the conclusion from all this? Does it improve as expected or not. Describe this, instead of just showing numbers in a table. The table is just there to backup the story.

p9l24 & Figure 9: This figure is inadequate. Again the location of the OMI pixels is not clear. Merge MODIS quicklookd into one RGB image and overlay the OMI pixel contours.

p10l3: the selection of the volcanic type is most probably.. : Most probably? Who is going to give a conclusive answer to that if not the authors themselves? First, indicate where the OMI pixels are in the MODIS RGB image as suggested above. Then, conclude whether or not this is due to the 'white area'... Do you mean cloud?

p10l10: perhaps indicating..: Again, why perhaps? Tell the reader whether there was dust or not. If not, why select this day? Surely a dust event can be easily found using OMI UVAI on a clear day. Indeed, 26 March 2008 shows low UVAI over the northern Sahara, so change this day and choose a day where you know what's going on and what aerosol model you should expect.

p10l24-26. So what's the conclusion here? Is the posterior uncertainty better or the same in the case of one chosen model? Does the (new) high uncertainty include the difference between the two measurements, or is it too small?

The conclusion section should be extended with a clear recommendation.

---

## Author Comment (AC1) · 14 Jul 2017

The authors thank Referee #1 for useful and thoughtful comments and suggestion to improve the paper. Below we answer to the comments point-by-point. The referee comments are in **bold**. The pages, lines and figures reported correspond to the manuscript under discussion.

**Response to Anonymous Referee #1**

The paper addresses a very important topic in aerosol retrievals from satellite: the uncertainty of the selected aerosol model and the impact on the AOD. Indeed, this is one of the largest uncertainties for aerosol AOD and the proposed method is very appropriate. It builds on current methods and elegantly generalises the techniques using a sound approach. The paper is well written and well structured and I feel this paper could give an important contribution to a more accurate retrieval of AOD from space based instruments.

There are a few improvement that I deem necessary for this paper to be acceptable. In its present form it lacks a clear definite conclusion and recommendation. A very decent physical and mathematical framework is presented, however at the end the reader is left with a somewhat unsatisfactory feeling, not knowing whether the whole exercise was successful or not. For me the questions that are addressed here are: 1) does the AOD retrieval improve when a combination of aerosol models is allowed and combined using the Bayesian model evidence? 2) Does the model selection uncertainty give a better estimate of the AOD uncertainty than the current one?

The authors pose the questions and address them, but I see no clear answer for these questions. It's left hanging in the conclusion section. It says 'the posterior probability distribution can characterise the uncertainty more extensively than commonly given standard deviation'. Fair enough, but what does this mean? Is it better? Should we generally apply this method? Also from the provided sensitivity studies it is just not clear whether things work as expected (probably leading to the general inconclusive conclusion section).

What I lack is an answer to these questions (supported by evidence): Does the average AOD perform better than the standard one, when compared to AERONET? If not, is this reflected in a larger uncertainty? If yes, are the AERONET and OMI AOD retrievals consistent within this new the uncertainty?

If this could be adequately answered, i recommend this paper for publication.

We thank the Referee #1 for reviewing our manuscript and encouraging general comments.

These are very important question. We have now included more discussion of these points in the end of Section 5 (Discussion and Conclusions). For the sake of clarity and as suggested by Anonymous Referee #2, we made also some other modifications in Section 5.

In order to highlight the main targets of our study, we added in the beginning of Conclusions section:

1) to improve the retrieval error estimate (i.e. to produce more realistic uncertainty estimate),

2) to evaluate the model choice procedure and

3) to find more robust AOD estimate that is based on the average of the most appropriate aerosol microphysical models instead of on a single model chosen probably by chance.

The aim was not actually to develop retrieval algorithm or improve the existing one (OMAERO), but to examine the influence of aerosol model selection to the resulted AOD and uncertainty.

It is difficult to give clear answers to the Referee #1 questions since we have not done yet the comprehensive testing and validation. However, we can present conclusions and give some recommendations based on the set of test cases done so far and based on the application of the method to measurements of one instrument.

**1) does the AOD retrieval improve when a combination of aerosol models is allowed and combined using the Bayesian model evidence? 2) Does the model selection uncertainty give a better estimate of the AOD uncertainty than the current one?**

1) In general, combination of aerosol models by utilizing Bayesian model averaging approach improve the retrieved AOD. From the test cases we can see that, usually, the averaged posterior gives better AOD estimate than if based on one best model, when compared to AERONET.

2) The uncertainty that accounts for the model selection has more information about the difficulty in model selection and thus the uncertainty is more realistic. We also considered the forward modeling uncertainty (i.e. model discrepancy) separately in order to take into account the imperfect forward modeling when fitting the LUT-based reflectance into observations.

**'the posterior probability distribution can characterise the uncertainty more extensively than commonly given standard deviation'. Fair enough, but what does this mean? Is it better?**

The posterior distributions of the best models and averaged posterior give more information about the uncertainty in model selection and in estimated AOD than one number and standard deviation can give. But presenting this pixel-wise uncertainty information, given by the posterior densities, in compact but still informative form is not clear. In addition, the other question could be what information of the uncertainty is needed or is sufficient to present. We have now removed this sentence since it was unclear statement (Discussion and Conclusions, p12l15-15) and added the following expression:.

"Moreover, further study and discussion is needed to determine how to express the uncertainty information, provided by the posterior distribution, in more compact form.

**Should we generally apply this method? Also from the provided sensitivity studies it is just not clear whether things work as expected**

The method brings more information about the uncertainty and it can be used for evaluation of the model selection process e.g. study the influence of aerosol microphysical model selection on the estimated AOD.

So far, the sensitivity studies have brought interesting information about the uncertainty related to the model selection process, e.g. difficulty in model selection or lack of appropriate model LUTs. However, the case studies have also shown that the aerosol type selection works as expected, albeit with some exceptions e.g. resulted unexpected type of model.

**Does the average AOD perform better than the standard one, when compared to AERONET? If not, is this reflected in a larger uncertainty? If yes, are the AERONET and OMI AOD retrievals consistent within this new the uncertainty?**

The case studies reveal that the proposed method using averaged AOD was not better than the standard one (OMAERO) if compared only the retrieved AOD values to AERONET. But the proposed method got solution for more pixels than OMAERO. Also the retrieved, but LUT-dependent, Ångström exponents were in rather good agreement with the AERONET values.

The test cases also show that, in general, the larger uncertainty, i.e. posterior width, reflected the uncertainty in the retrieval. Also, when the deviation from the AERONET AOD was larger then the uncertainty was higher.

**Minor comments:**

p1l24 (and a few more): data is -> data are
Corrected

**p3l25: referred -> referred to** Corrected.

**p3l28: a cloudy ground pixel sounds strange. I would say a cloudy atmosphere pixel. Or just a cloudy pixel.**

We changed "a cloudy ground pixel" to "a cloudy pixel".

**p3l31: What is a wise quality?**

This is a typo; "wise" removed

**p4l4: Before the start of the new sentence, add 'For surface reflectivity,' (we used.. etc)**

Added, thank you.

**p4l11-13: you say: the band at 477 nm adds important info, yet you exclude it specifically. Why?**

We excluded band at 477 nm since we did not need aerosol layer height information in our study. The other reason is based on experimental issue since we found that this band brought extra complexity when examined the modeled spectral reflectance fit to the observed reflectance.

We changed the order of the last two sentences in the revised manuscript and rephrased the sentence as:

"However, we omitted in our study the band 477 nm due to experimental purpose and since we did not need aerosol height information."

**p5l3: Equation (1) is not just a 'formula'. Start this discussion with a physical description like: Assuming a Lambertian surface the contribution of the radiation at the TOA can be separated from that of the atmosphere (e,g. Chandrasekhar, 1960), viz. etc.**

Thank you, we have rephrased the sentence as suggested. The sentence now reads:

"Assuming a Lambertian surface the contribution of the radiation at the TOA can be separated from that of the atmosphere (e.g. Chandrasekhar, 1960) leading to the equation for modeled reflectance as ..."

**p5l10: of the real -> of the aerosols in the real**

Corrected.

**p5l10-11: This forward model app error,.. Which one? You haven't described an error yet. Do you mean the difference between real and approx. reflectances? Then describe that.**

By "forward model approximation error" we mean error that originates from forward model approximation. The beginning of the sentence has been revised to "Approximations in forward modeling ..."

**p5l13: This is strange: I would expect that a total (megs) error would be forward model error, noise (and perhaps more). Noise surely doesn't include forward model error? What is epsilon\_obs? Noise or total? Rephrase l11.**

Thank you for notifying this incoherent statement; "Measurement noise" is a wrong expression.

We have now removed this sentence since it is unnecessary here.

 $\varepsilon_{obs}$  is the measurement error (or noise) and  $\varepsilon_{obs}(\lambda) \sim N(0, \sigma^2_{obs}(\lambda))$ . We have clarified this in the revised manuscript.

To make clear, we have also expressed the measurement error standard deviation  $\sigma_{obs}(\lambda) = R_{obs}(\lambda)/SNR$  (in p5l20) and changed the notation " $\sigma(\lambda)$ " to " $\sigma_{obs}(\lambda)$ " (in Eq. 3)

**p6. Increase the size of eq. 4 and 5, like eq. 1. They are the basis of the paper.**

Done.

**p7l19: cover -> covers** Corrected.

**p7l20: cover -> covers a** Corrected.

**p7l22-25: Move this to section 2. And add a description of MODIS, which is introduced in the next paragraph.**

The text part p7l22-25 introduces the AERONET data that are used for evaluating the case studies. That's why we would like to retain the AERONET description part in this Section 4 (Case studies and results).

As suggested, we have now added a description of MODIS in the end of Section 4 (p7). The added text reads:

"We tracked clouds and land scene for the case studies by utilizing true-color images from MODIS, on Aqua satellite, that has the equator crossing time only about 15 minutes earlier than OMI. The MODIS instrument is onboard both Terra and Aqua spacecraft. The data products derived from MODIS measurements include atmosphere (e.g. cloud mask and aerosol products), land, cryosphere and ocean products (see e.g. http://modis.gsfc.nasa.gov)."

**p8l1 & Figure 1. The OMI pixels -> The OMI pixels that were analysed The location of the OMI pixels within the MODIS swath are not clear. In Figure 1 add the contours of the OMI pixels that are used in Fig 2-6.**

P8l1 Corrected: "The OMI pixels" -> "The OMI pixels that were analysed". Fig 1.: We added contours of the OMI pixels.

We also added in the figure caption: "The area of analysed OMI pixels is marked with red contours."

**p8l2: The pixel has no data if -> No data are reported if the pixel is** Corrected**

**p8l18: pixel wise -> pixel-wise** Corrected.

**p823: in the latter day case -> On the 27th,** Corrected.

**p9l8-10: Figure 8 is superfluous. Remove it and on describe the results from it in the text. It will reduce the number of figures, which is needed anyway.**

We agree. We removed Fig 8., rephrased the text accordingly and moved it to the beginning of the paragraph. The sentence reads now:

"The other types of models, e.g. weakly absorbing type, do not match as well as the selected best BB models."

**p9l11-13;: Elaborate on this result. It is as important as the 16th.**

As suggested, we have now included more discussion in the revised manuscript about the results of 27th case.

**p9l14-22: Here's the first missing conclusion. So you compared the Angstrom exponents. Whats the conclusion from all this? Does it improve as expected or not. Describe this, instead of just showing numbers in a table. The table is just there to backup the story.**

The conclusion from comparison with AERONET values is that the derived, even if LUT dependent, Ångström exponent values are in rather good agreement with the AERONET values (see Table 2). But in some cases we observed that agreement between the AOD values do not necessarily lead to agreement between the Ångström exponent values.

Unfortunately we cannot answer to the question "does it improve as expected or not" since we have only done the comparison between LUT-based derived  $\alpha 1(442-500 \text{ nm})$  (or  $\alpha 2$ ) and AERONET  $\alpha (440-675 \text{ nm})$ .

We have now included the following description in the end of section 4.1 in the revised version:

"For Beijing case, in both days, the derived Ångström exponent value of the best model ( $\alpha$ 1) is in good agreement with the AERONET value. Even so, on the 16th of April  $\alpha$ 2 deviates more from the AERONET value although the estimated AOD, based on the second best model, is closer to the AERONET AOD values (see Fig. 7 left)."

For Africa case, the Figure 14 shows distribution of  $\alpha 1$  (left) and  $\alpha 2$  (right) values, respectively. We added the following description in the revised manuscript (p10):

"That is, the Ångström values are low where the desert dust type of models dominate. Correspondingly, in the coastal region where typically is smoke and urban polluted air the Ångström exponent is higher."

When we compare Ångström exponent values at locations of AERONET sites in Africa the agreement is generally good, except at DMN\_Maine\_Soroa site.

We added the following sentences related to Ångström exponent comparison (p10) for Agoufou:

"However, the derived Ångström exponent  $\alpha 1$  has rather good agreement with the AERONET value (Table 2)."

and for DMN\_Maine\_Soroa:

"... but the derived Ångström exponents,  $\alpha 1$  and  $\alpha 2$ , do not agree with the AERONET value (Table 2)."

and for IER\_Cinzana and Saada:

"For the sites IER\_Cinzana and Saada the best and the second best models have as good evidence (Fig. 15 right column) indicating that the selection of the best model happened by chance. Consequently, the derived  $\alpha 1$  for Saada site is consistent with the AERONET value whereas the derived  $\alpha 2$  for IER\_Cinzana has better agreement than  $\alpha 1$  with the AERONET value."

**p9124 & Figure 9: This figure is inadequate. Again the location of the OMI pixels is not clear. Merge MODIS quicklookd into one RGB image and overlay the OMI pixel contours.**

We have now merger the two MODIS RGB images and added the contours of the OMI pixels. We also added in the figure caption: "The area of analysed OMI pixels is marked with red contours."

**p10l3: the selection of the volcanic type is most probably.. : Most probably? Who is going to give a conclusive answer to that if not the authors themselves? First, indicate where the OMI pixels are in the MODIS RGB image as suggested above. Then, conclude whether or not this is due to the 'white area'... Do you mean cloud?**

The contours of the OMI pixels added in the MODIS RGB image as suggested above.

We have now changed the notation "white area" as "cloud" and rephrased the text as:

"The selection of volcanic aerosol type as the only appropriate aerosol type happens for pixels located northeast from the Lake Chad where is seen cloud in the MODIS RGB image (Fig. 8)."

p10l10: perhaps indicating..: Again, why perhaps? Tell the reader whether there was dust or not. If not, why select this day? Surely a dust event can be easily found using OMI UVAI on a clear day. Indeed, 26 March 2008 shows low UVAI over the northern Sahara, so change this day and choose a day where you know what's going on and what aerosol model you should expect.

The criterion for selecting that date, 26 March, is almost cloud free scene over Northern and Central Africa thus providing large pixel area to study. The aim was to study the uncertainty in aerosol model selection and its effect on the results in "the normal aerosol situation" and we did not seek a special case with known dust or smoke event. The resulted aerosol types were what we expected i.e. dust in the north and urban pollution/smoke in the coast region.

We have now removed this imprecise statement (p10l10):

"The retrieved AOD estimates are rather small perhaps indicating that no dust event or active fires were going on."

in the manuscript since it is not relevant here.

**p10l24-26. So what's the conclusion here? Is the posterior uncertainty better or the same in the case of one chosen model? Does the (new) high uncertainty include the difference between the two measurements, or is it too small?**

The paragraph p10l24-26 considers results in one OMI pixel located around AERONET Agoufou site.

As a result there is only one selected model having a sufficient, even poor, fit to the measured reflectance. As expected, the large width of the posterior distribution, that is the averaged posterior distribution as well, indicates high uncertainty in the model selection and thus in the retrieved AOD. Consequently, the answer to the first question is: even the method gives a solution that passed the goodness-of-fit test it does not ensure correctness of the result.

The answer to the second questions is: the posterior uncertainty is the same in case of one chosen model.

The retrieval uncertainty is high and still the posterior density does not cover the AERONET Agoufou AOD values (or daily average) but it covers the OMAERO AOD (1.557). However, the Ångström exponent values for AERONET daily average  $\alpha(440-675 \text{ nm})$  and proposed method  $\alpha 1(442-500 \text{ nm})$ , i.e. 0.375 and 0.293 respectively, match quite well (Table 2).

It must be noted here that the AERONET measurements at the Agoufou site were made in the morning and the last one about 3.5 hours before OMI overpass time.

We have now added more discussion and rephrased the paragraph p10l24-28 in the revised version.

**The conclusion section should be extended with a clear recommendation.**

We have included more text for recommendation, and hopefully in clear way, in the Discussion and Conclusions section.

---

## Author Response (AR1)

Dear Associate Editor,

Please find below our detailed point-by-point response to all Anonymous Referee #1 and Referee #2 comments and specification of all changes in the revised manuscript. We have also attached an Appendix as a supplement material to the Referee #2 response. The marked-up manuscript version showing the changes made is given in the end.

The response to the Referees is structured as follows: (1) comments from Referee in bold (2) author's response in normal font and (3) author's changes in manuscript in blue.

Please note that the page, line and figure numbers in the given response and changes description refer here to the marked-up manuscript.

We like to note that the main changes in the text concern Section 5 (Discussion and Conclusions).

With best regards, Anu Kauppi Finnish Meteorological Institute

**Response to Anonymous Referee #1**

There are a few improvement that I deem necessary for this paper to be acceptable. In its present form it lacks a clear definite conclusion and recommendation. A very decent physical and mathematical framework is presented, however at the end the reader is left with a somewhat unsatisfactory feeling, not knowing whether the whole exercise was successful or not. For me the questions that are addressed here are: 1) does the AOD retrieval improve when a combination of aerosol models is allowed and combined using the Bayesian model evidence? 2) Does the model selection uncertainty give a better estimate of the AOD uncertainty than the current one?

The authors pose the questions and address them, but I see no clear answer for these questions. It's left hanging in the conclusion section. It says 'the posterior probability distribution can characterise the uncertainty more extensively than commonly given standard deviation'. Fair enough, but what does this mean? Is it better? Should we generally apply this method? Also from the provided sensitivity studies it is just not clear whether things work as expected (probably leading to the general inconclusive conclusion section).

What I lack is an answer to these questions (supported by evidence): Does the average AOD perform better than the standard one, when compared to AERONET? If not, is this reflected in a larger uncertainty? If yes, are the

**AERONET and OMI AOD retrievals consistent within this new the uncertainty?**

**If this could be adequately answered, i recommend this paper for publication.**

These are very important question. We have now included more discussion of these points in the end of Section 5 (Discussion and Conclusions) (p12-14). For the sake of clarity and as suggested by Anonymous Referee #2, we made also some other modifications in Section 5.

In order to highlight the main targets of our study, we added in the beginning of Conclusions section (p12 lines13-16):

1) to improve the retrieval error estimate (i.e. to produce more realistic uncertainty estimate),

2) to evaluate the model choice procedure and

3) to find more robust AOD estimate that is based on the average of the most appropriate aerosol microphysical models instead of on a single model chosen probably by chance.

The aim was not actually to develop retrieval algorithm or improve the existing one (OMAERO), but to examine the influence of aerosol model selection to the resulted AOD and uncertainty.

It is difficult to give clear answers to the Referee #1 questions since we have not done yet the comprehensive testing and validation. However, we can present conclusions and give some recommendations based on the set of test cases done so far and based on the application of the method to measurements of one instrument.

**1) does the AOD retrieval improve when a combination of aerosol models is allowed and combined using the Bayesian model evidence? 2) Does the model selection uncertainty give a better estimate of the AOD uncertainty than the current one?**

1) In general, combination of aerosol models by utilizing Bayesian model averaging approach improve the retrieved AOD. From the test cases we can see that, usually, the averaged posterior gives better AOD estimate than if based on one best model, when compared to AERONET.

We have added discussion of this in the conclusions section (p13 lines33-35 and p14 lines1-2)

2) The uncertainty that accounts for the model selection has more information about the difficulty in model selection and thus the uncertainty is more realistic.

We also considered the forward modeling uncertainty (i.e. model discrepancy) separately in order to take into account the imperfect forward modeling when fitting the LUT-based reflectance into observations.

We have added discussion of this in the conclusions section (p13 lines25-33)

**'the posterior probability distribution can characterise the uncertainty more extensively than commonly given standard deviation'. Fair enough, but what does this mean? Is it better?**

The posterior distributions of the best models and averaged posterior give more information about the uncertainty in model selection and in estimated AOD than one number and standard deviation can give. But presenting this pixel-wise uncertainty information, given by the posterior densities, in compact but still informative form is not clear. In addition, the other question could be what information of the uncertainty is needed or is sufficient to present.

We have now removed this sentence since it was unclear statement (Discussion and Conclusions, p13 lines31-32) and added the following expression (p14 lines11-12):.

"Moreover, further study and discussion is needed to determine how to express the uncertainty information, provided by the posterior distribution, in more compact form.

**Should we generally apply this method? Also from the provided sensitivity studies it is just not clear whether things work as expected**

The method brings more information about the uncertainty and it can be used for evaluation of the model selection process e.g. study the influence of aerosol microphysical model selection on the estimated AOD.

So far, the sensitivity studies have brought interesting information about the uncertainty related to the model selection process, e.g. difficulty in model selection or lack of appropriate model LUTs. However, the case studies have also shown that the aerosol type selection works as expected, albeit with some exceptions e.g. resulted unexpected type of model.

We have added discussion of these issues in the conclusions section (p13 lines25-35 and p14 lines1-2, p14 lines8-12)

**Does the average AOD perform better than the standard one, when compared to AERONET? If not, is this reflected in a larger uncertainty? If yes, are the AERONET and OMI AOD retrievals consistent within this new the uncertainty?**

The case studies reveal that the proposed method using averaged AOD was not better than the standard one (OMAERO) if compared only the retrieved AOD values to AERONET. But the proposed method got solution for more pixels than OMAERO. Also the retrieved, but LUT-dependent, Ångström exponents were in rather good agreement with the AERONET values.

The test cases also show that, in general, the larger uncertainty, i.e. posterior width, reflected the uncertainty in the retrieval. Also, when the deviation from the AERONET AOD was larger then the uncertainty was higher.

We have added discussion of this in the conclusions section (p14 lines3-7)

**Minor comments:**

p1l24 (and a few more): data is -> data are Corrected (p1line24, p2line16 and p14line18)

**p3l25: referred -> referred to**

Corrected. (p3 line27)

p3l28: a cloudy ground pixel sounds strange. I would say a cloudy atmosphere pixel. Or just a cloudy pixel.

We changed "a cloudy ground pixel" to "a cloudy pixel". (p3 line30)

**p3l31: What is a wise quality?**

This is a typo; "wise" removed (p4 line1)

**p4l4: Before the start of the new sentence, add 'For surface reflectivity,' (we used.. etc)**

Added, thank you. (p4 line6)

**p4l11-13: you say: the band at 477 nm adds important info, yet you exclude it specifically. Why?**

We excluded band at 477 nm since we did not need aerosol layer height information in our study. The other reason is based on experimental issue since we found that this band brought extra complexity when examined the modeled spectral reflectance fit to the observed reflectance.

We changed the order of the last two sentences in the revised manuscript and rephrased the sentence as (p4 lines13-16):

"However, we omitted in our study the band 477 nm due to experimental purpose and since we did not need aerosol height information."

**p5l3: Equation (1) is not just a 'formula'. Start this discussion with a physical description like: Assuming a Lambertian surface the contribution of the radiation at the TOA can be separated from that of the atmosphere (e,g. Chandrasekhar, 1960), viz. etc.**

Thank you, we have rephrased the sentence as suggested.

The sentence now reads (p5 lines5-7):

"Assuming a Lambertian surface the contribution of the radiation at the TOA can be separated from that of the atmosphere (e.g. Chandrasekhar, 1960) leading to the equation for modeled reflectance as ..."

**p5l10: of the real -> of the aerosols in the real**

Corrected. (p5 line15)

**p5l10-11: This forward model app error,.. Which one? You haven't described an error yet. Do you mean the difference between real and approx. reflectances? Then describe that.**

By "forward model approximation error" we mean error that originates from forward model approximation.

The beginning of the sentence has been revised to "Approximations in forward modeling ..." (p5 line16)

p5l13: This is strange: I would expect that a total (megs) error would be forward model error, noise (and perhaps more). Noise surely doesn't

**include forward model error? What is epsilon\_obs? Noise or total? Rephrase l11.**

Thank you for notifying this incoherent statement; "Measurement noise" is a wrong expression.

We have now removed this sentence since it is unnecessary here. (p5 line19)

 $\epsilon_{obs}$  is the measurement error (or noise) and  $\epsilon_{obs}(\lambda) \sim N(0, \sigma^2_{obs}(\lambda))$ . We have clarified this in the revised manuscript. (p5 line24)

To make clear, we have also expressed the measurement error standard deviation  $\sigma_{obs}(\lambda) = R_{obs}(\lambda)/SNR$  (in p5 line27) and changed the notation " $\sigma(\lambda)$ " to " $\sigma_{obs}(\lambda)$ " (in Eq. 3)

p6. Increase the size of eq. 4 and 5, like eq. 1. They are the basis of the paper. Done. (p6 lines 9 & 17)

**p7l19: cover -> covers** Corrected. (p8 line5)

p7l20: cover -> covers a Corrected. (p8 line6)

**p7l22-25: Move this to section 2. And add a description of MODIS, which is introduced in the next paragraph.**

The text part p7l22-25 introduces the AERONET data that are used for evaluating the case studies. That's why we would like to retain the AERONET description part in this Section 4 (Case studies and results)(p8 lines8-11).

As suggested, we have now added a description of MODIS in the end of Section 4. The added text reads (p8 lines11-14):

"We tracked clouds and land scene for the case studies by utilizing true-color images from MODIS, on Aqua satellite, that has the equator crossing time only about 15 minutes earlier than OMI. The MODIS instrument is onboard both Terra and Aqua spacecraft. The data products derived from MODIS measurements include atmosphere (e.g. cloud mask and aerosol products), land, cryosphere and ocean products (see e.g. http://modis.gsfc.nasa.gov)."

p8l1 & Figure 1. The OMI pixels -> The OMI pixels that were analysed The location of the OMI pixels within the MODIS swath are not clear. In Figure 1 add the contours of the OMI pixels that are used in Fig 2-6.

(P8 line20) Corrected: "The OMI pixels" -> "The OMI pixels that were analysed". Fig 1.: We added contours of the OMI pixels.

We also added in the figure caption: "The area of analysed OMI pixels is marked with red contours." (p18)

p8l2: The pixel has no data if -> No data are reported if the pixel is Corrected (p8 lines21-22)

**p8l18: pixel wise -> pixel-wise**

Corrected. (p9 line3)

p823: in the latter day case -> On the 27th, Corrected. (p9 line8)

**p9l8-10: Figure 8 is superfluous. Remove it and on describe the results from it in the text. It will reduce the number of figures, which is needed anyway.**

We agree. We removed Fig 8. (p9 lines29-31), rephrased the text accordingly and moved it to the beginning of the paragraph. The sentence reads now (p9 lines23-24):

"The other types of models, e.g. weakly absorbing type, do not match as well as the selected best BB models."

**p9l11-13;: Elaborate on this result. It is as important as the 16th.**

As suggested, we have now included more discussion in the revised manuscript about the results of 27th case. (p9 lines32-35 and p10 lines1-5)

**p9l14-22: Here's the first missing conclusion. So you compared the Angstrom exponents. Whats the conclusion from all this? Does it improve as expected or not. Describe this, instead of just showing numbers in a table. The table is just there to backup the story.**

The conclusion from comparison with AERONET values is that the derived, even if LUT dependent, Ångström exponent values are in rather good agreement with the AERONET values (see Table 2). But in some cases we observed that agreement between the AOD values do not necessarily lead to agreement between the Ångström exponent values.

Unfortunately we cannot answer to the question "does it improve as expected or not" since we have only done the comparison between LUT-based derived  $\alpha 1(442-500 \text{ nm})$  (or  $\alpha 2$ ) and AERONET  $\alpha (440-675 \text{ nm})$ .

We have now included the following description in the end of section 4.1 in the revised version (p10 lines14-17):

"For Beijing case, in both days, the derived Ångström exponent value of the best model ( $\alpha$ 1) is in good agreement with the AERONET value. Even so, on the 16th of April  $\alpha$ 2 deviates more from the AERONET value although the estimated AOD, based on the second best model, is closer to the AERONET AOD values (see Fig. 7 left)."

For Africa case, the Figure 13 shows distribution of  $\alpha 1$  (left) and  $\alpha 2$  (right) values, respectively. We added the following description in the revised manuscript (p11 lines14-16):

"That is, the Ångström values are low where the desert dust type of models dominate. Correspondingly, in the coastal region where typically is smoke and urban polluted air the Ångström exponent is higher."

When we compare Ångström exponent values at locations of AERONET sites in Africa the agreement is generally good, except at DMN\_Maine\_Soroa site.

We added the following sentences related to Ångström exponent comparison for Agoufou (p11 lines30-31):

"However, the derived Ångström exponent  $\alpha 1$  has rather good agreement with the AERONET value (Table 2)."

**and for DMN\_Maine\_Soroa (p11 line35):**

"... but the derived Ångström exponents,  $\alpha 1$  and  $\alpha 2$ , do not agree with the AERONET value (Table 2)."

and for IER\_Cinzana and Saada (p12 lines5-9):

"For the sites IER\_Cinzana and Saada the best and the second best models have as good evidence (Fig. 15 right column) indicating that the selection of the best model happened by chance. Consequently, the derived  $\alpha 1$  for Saada site is consistent with the AERONET value whereas the derived  $\alpha 2$  for IER\_Cinzana has better agreement than  $\alpha 1$  with the AERONET value."

**p9124 & Figure 9: This figure is inadequate. Again the location of the OMI pixels is not clear. Merge MODIS quicklookd into one RGB image and overlay the OMI pixel contours.**

We have now merger the two MODIS RGB images (figure 8) and added the contours of the OMI pixels. We also added in the figure caption (p24): "The area of analysed OMI pixels is marked with red contours."

p10l3: the selection of the volcanic type is most probably.. : Most probably? Who is going to give a conclusive answer to that if not the authors themselves? First, indicate where the OMI pixels are in the MODIS RGB image as suggested above. Then, conclude whether or not this is due to the 'white area'... Do you mean cloud?

The contours of the OMI pixels added in the MODIS RGB image as suggested above. (fig.8 p24)

We have now changed the notation "white area" as "cloud" and rephrased the text as (p10 lines30-33):

"The selection of volcanic aerosol type as the only appropriate aerosol type happens for pixels located northeast from the Lake Chad where is seen cloud in the MODIS RGB image (Fig. 8)."

p10l10: perhaps indicating..: Again, why perhaps? Tell the reader whether there was dust or not. If not, why select this day? Surely a dust event can be easily found using OMI UVAI on a clear day. Indeed, 26 March 2008 shows low UVAI over the northern Sahara, so change this day and choose a day where you know what's going on and what aerosol model you should expect.

The criterion for selecting that date, 26 March, is almost cloud free scene over Northern and Central Africa thus providing large pixel area to study. The aim was to study the uncertainty in aerosol model selection and its effect on the results in "the normal aerosol situation" and we did not seek a special case with known dust or smoke event. The resulted aerosol types were what we expected i.e. dust in the north and urban pollution/smoke in the coast region. We have now removed this imprecise statement (p11 line6): "The retrieved AOD estimates are rather small perhaps indicating that no dust event or active fires were going on." in the manuscript since it is not relevant here.

**p10l24-26. So what's the conclusion here? Is the posterior uncertainty better or the same in the case of one chosen model? Does the (new) high uncertainty include the difference between the two measurements, or is it too small?**

The paragraph p10l24-26 considers results in one OMI pixel located around AERONET Agoufou site.

As a result there is only one selected model having a sufficient, even poor, fit to the measured reflectance. As expected, the large width of the posterior distribution, that is the averaged posterior distribution as well, indicates high uncertainty in the model selection and thus in the retrieved AOD. Consequently, the answer to the first question is: even the method gives a solution that passed the goodness-of-fit test it does not ensure correctness of the result.

The answer to the second questions is: the posterior uncertainty is the same in case of one chosen model.

The retrieval uncertainty is high and still the posterior density does not cover the AERONET Agoufou AOD values (or daily average) but it covers the OMAERO AOD (1.557). However, the Ångström exponent values for AERONET daily average  $\alpha(440-675 \text{ nm})$  and proposed method  $\alpha 1(442-500 \text{ nm})$ , i.e. 0.375 and 0.293 respectively, match quite well (Table 2).

It must be noted here that the AERONET measurements at the Agoufou site were made in the morning and the last one about 3.5 hours before OMI overpass time.

We have now added more discussion and rephrased the paragraph in the revised version. (p11 lines23-30)

**The conclusion section should be extended with a clear recommendation.** We have included more text for recommendation, and hopefully in clear way, in the Discussion and Conclusions section. (p13 lines25-28 and p14 lines8-12)

**Response to Anonymous Referee #2**

**The paper is interesting with sound math. For it to be published and attract wider readability, it needs significant revisions, especially in many places where Maatta et al's paper is referred.**

We agree that we have frequently referred to the paper Määttä et al. (2014) for the theoretical details. In that paper the theoretical background and construction of the method is presented in detail and we did not want to repeat it in this manuscript.

As suggested by the Referee #2, we have now included more description of the method in the revised version and took into account the detailed Referee

comments listed below. We have also included an Appendix document, as supplement to the comments, for describing our computational implementation of the method.

**Detailed comments are**

**1. what is the key difference in method between this paper and Maatta et al? Is it simply that Maatta et al. didn't analyze the retrieval uncertainty (p. 11, line 5)?**

The method is the same in both papers and applied to OMI measurements. Both papers also analyze the retrieval uncertainty.

The difference is that in the paper Määttä et al. the test cases examined the method at single OMI pixels using two ways: with and without the included model discrepancy term. Whereas in this manuscript the method includes the model discrepancy term and is applied to more comprehensive data set. In addition this manuscript considers more the presentation and description of the uncertainty due to the model selection.

We have now rephrases the first part of Section 5 (Discussion and Conclusions) to express more clearly the difference between the papers. (p12 lines18-21)

**2. equation 1. To compute reflectance, one needs to know path reflectance that in turn is related to aerosol optical depth. the same is true for transmittance. Please explain how the calculation in equation 1 is implemented? what are the inputs and from where?**

The path reflectance  $R_a(\lambda,\tau,\mu,\mu_0,\Delta\phi,ps)$  and transmittance  $T(\lambda,\tau,\mu,\mu_0,p_s)$  are both related to  $\tau$  (i.e. AOD). They, as well as spherical albedo  $s(\lambda,\tau,p_s)$ , are taken from the associated multi-dimensional model table LUT by interpolating between LUT contained nodal point values of  $\tau$ ,  $\Delta\phi$ ,  $p_s$ ,  $\mu$  and  $\mu_0$ . This is explained in the manuscript (Discussion paper: page 5 lines 1-7).

As input data we use wavelength bands  $\lambda$  and sun-satellite geometry data included in OMI data ( $\Delta \phi$ ,  $p_s$ ,  $\mu$  and  $\mu_0$ ). Please see Section 2.1. and e.g. Torres et al., (2002) for more information about content of OMI LUTs.

We added in the revised manuscript the sentence (page 5): "The sun-satellite geometry data  $\Delta \phi$ , ps,  $\mu$  and  $\mu_0$  are included in the OMI Level 1B data." (p5 lines11-12). We also changed "by LUT" to "from LUT" for clarity (p5 line 9).

**3. equation 2. Is observation error kept constant for each wavelength in this case?**

The observation error  $\varepsilon_{obs}(\lambda)$  is assumed to be Gaussian distributed with zero mean and variance  $\sigma^2_{obs}(\lambda)$ . The variance is not constant as the standard deviation is calculated by  $\sigma_{obs}(\lambda) = R_{obs}(\lambda)/SNR$  where we set SNR=500. To make this clear we added  $\varepsilon_{obs}(\lambda) \sim N(0, \sigma^2_{obs}(\lambda))$  and  $\sigma_{obs}(\lambda) = R_{obs}(\lambda)/SNR$  in the revised manuscript in Section 3.1. (p5 line24 and line27)

4. page 5. line 25, "we constructed the covariance function empirically by using the wavelength distance dependent correlation structure of the residuals (See Maata et al 2014 details)". This sentence is very difficult to understand. The paper should standalone by itself.

We have now clarified this sentence in the revised manuscript. We also changed the "covariance function" to "covariance matrix" for simplicity. However, the theoretical details are left to the reference paper Määttä et al. (2014).

In hope of clarifying the process for constructing the covariance matrix C, we have changed the sentence to:

"The covariance matrix C was constructed by means of an empirical semivariogram when the variances of the residual differences were calculated for each wavelength pairs with the distance d. Next, the theoretical Gaussian variogram model was fitted to these empirical semivariogram values. The outcome of this analysis were the values for parameters that defines the model discrepancy covariance matrix C (see Määttä et al. (2014) for details)." (p6 lines2-6)

**5.** eq. **3.** Where does this equation come from? how is measurement error variance computed?**

Equation (3) is the likelihood function that describes the distribution of the observations given the model and is dependent on the residuals. The likelihood has that form (Eq. 3) since we assume it follows a multivariate Gaussian distribution with non-diagonal covariance matrix C+diag( $\sigma^2_{obs}(\lambda)$ ). Here C is the model discrepancy covariance matrix.

We added the following sentence in the revised manuscript (page 5 line 28): "We assume that the likelihood function describing the distribution of the observations given the model follows a Gaussian distribution." (p6 lines7-8)

The measurement error variance  $\sigma^2_{obs}(\lambda)$  is computed as described above (the comment 3), i.e.  $\sigma^2_{obs}(\lambda) = (R_{obs}(\lambda)/SNR)^2$  where we have used SNR = 500. Formula for  $\sigma_{obs}(\lambda)$  added (p5 line 27)

**6. eq. 4. It is not clear how p(tau|m) is constructed. "In the present case, the estimation and model selection procedure seeks the solution for a onedimensional parameters tau, and the calculations will be fairly straightforward by numerical quadrature. The posterior distribution calculation is presented in the more detail in Maata et al 2014". Again, this reviewer doesn't understand this.**

The prior  $p(\tau|m)$ , i.e. the prior distribution for  $\tau$  depending on the aerosol microphysical model m, is constructed in our study by assuming that it follows a log-normal distribution with mean value, say 2. This confirms that  $p(\tau|m)$  can take only positive real values and thus ensures that estimated AOD is positive.

We have now added in the revised manuscript the sentence (page 6 line 19-20): "We assumed that the prior  $p(\tau|m)$  follows a log-normal distribution in order to ensure that the estimated AOD is positive."

This sentence was unclear and we have now rephrased the text in the revised manuscript as (p6 lines 21-24):

"In our case, the model selection procedure seeks the solution for one parameter  $\tau$  and then the calculation of posterior distribution is fairly straightforward by numerical quadrature. The calculation of the posterior distribution is presented in more detail in Määttä et al. (2014)."

Please, note that we have also included an Appendix document, as supplement to the comments, for describing our computational implementation of the method.

**7. P6, L11-15. How the evidence is computed? This reviewer doesn't understand this paragraph. Later again, Maata et al 2014 is cited, generating a pause in text flow.**

The evidence  $p(R_{obs}|m)$  is calculated by numerical integration  $p(R_{obs}|m) = \int p(R_{obs}|\tau,m) p(\tau|m) d\tau$ .

We added this formula and the sentence reads now (p6 lines25-26)

"The denominator  $p(R_{obs}|m) = \int p(R_{obs}|\tau, m) p(\tau|m) d\tau$  in Eq. (4) is the probability of the observed reflectance Robs assuming the model m is the correct one."

**8. Overall, the method presented here lacks materials for readers to comprehend. An specific example will be helpful to illustrate how all these equations are implemented.**

We have now included Appendix document, as supplement to the Referee #2 comments, for describing computational implementation of the method.

**9. The results part also lack validation or inter-comaprison with MODIS AOD. Does the method help to interpret the inter-comparison?**

The inter-comparison with MODIS, or with other satellite retrievals, is outside scope of this paper.

This is an interesting question. The method presented can help to interpret the results from inter-comparison if the uncertainty is determined and characterized in a way it is comparable. But in principle, maybe in the future, the method described can give additional benefit for the inter-comparison.

**10.** Introduction part needs to include couple of references that reflect the research activities in U.S.**

(1) p2, L15. The following paper used AOD to constrain the emissions as well.

Wang, J. et al., 2012. Top-down estimate of dust emissions through integration of MODIS and MISR aerosol retrievals with the GEOS-Chem adjoint model, Geophys. Res. Lett. L08802.

Xu et al., 2013. Constraints on aerosol sources using GEOS-Chem adjoint and MODIS radiances, and evaluation with Multi-sensor (OMI, MISR) data, J. Geophys. Res. At- mos., 118, 6396-6413.

We have now added the reference papers Wang et al. (2012) and Xu et al. (2013). We also rephrased the sentence in as (p2 lines14-16)

"Furthermore, the satellite based data can be combined with numerical models when estimating aerosol emission fluxes (Huneeus et al., 2012) or spatially constraining amount of aerosol emissions (Wang et al., 2012; Xu et al., 2013)."

**(2) P2, L 17-27. while LUT is widely used in operational retrieval algorithm, several research algorithm used aerosol properties from chemistry transport models. This point should be mentioned here.**

We apologizes we do not have better knowledge of the research algorithms that use chemistry transport models for aerosol properties. And we do not have a reference to this research work either.

But in order to mention the use of climate models in the retrievals we have added the following sentence (p2 lines23-24):

"The aerosol properties in the LUTs can be based on observations or combination of observations and climate models (Holzer-Popp et al., 2013)."

**A Appendix**

**A.1 Computational implementation of the method**

This Appendix presents a pseudo-code for implementation of a method applied in manuscript Kauppi et al. (2017) and introduced in paper Määttä et al. (2014) step-by-step for a one Ozone Monitoring Instrument (OMI) pixel. The method is based on Bayesian inference approach.

OMI Data:

- The observed top-of-the-atmosphere (TOA) spectral reflectance  $\vec{R}_{obs}(\lambda)$  at selected wavelength bands  $\lambda = (\lambda_1, \ldots, \lambda_n)$  calculated from the OMI Level 1B VIS and UV radiances and Level 1B Solar irradiance data
- The measurement error variances  $\sigma^2_{\rm obs}(\lambda), \lambda = (\lambda_1, \dots, \lambda_n)$
- The set of Look-up-tables (LUTs) containing pre-calculated aerosol microphysical models (e.g. hdf5 files)

Outcome:

- Posterior distribution  $p(\tau | \vec{R}_{obs}, m)$  of  $\tau$  (i.e. AOD) given as a discrete set of values for  $\tau$  in the range of  $[0, \tau_{max}]$ . The posterior distribution is evaluated for each selected best fitting model (maximum of 10) and stored in a table.
- Averaged posterior distribution  $p_{\text{avg}}(\tau | \vec{R}_{\text{obs}})$  given as a discrete set of values for  $\tau$  in the range of  $[0, \tau_{\text{max}}]$  and stored in a table.
- Point estimate for AOD at 500 nm determined as maximum a posteriori (MAP) estimate, i.e. mode of the averaged posterior distribution

We use a symbol  $\tau$  for AOD in the formulas. The modeled reflectance  $\vec{R}_{\rm mod}(\tau, \lambda)$  depends on  $\tau$  and is calculated by interpolation between nodal values of LUT while fitted to the measured reflectance  $\vec{R}_{\rm obs}$  in order to find  $\tau$  that minimizes

$$\chi^2_{\rm mod}(\tau) = \vec{R}_{\rm res}(\lambda)^T \left(\mathbf{C} + \operatorname{diag}(\sigma^2_{\rm obs}(\lambda))\right)^{-1} \vec{R}_{\rm res}(\lambda).$$
(1)

Here  $\vec{R}_{\rm res}(\lambda) = \vec{R}_{\rm obs}(\lambda) - \vec{R}_{\rm mod}(\tau, \lambda)$  is the residual of model fit. This is done for each aerosol microphysical model in turn. In the formula  $\sigma_{\rm obs}^2(\lambda)$  are the measurement error variances and **C** is non-diagonal covariance matrix for model discrepancy (i.e. forward modelling uncertainty). In our experiment we calculated the elements of the covariance matrix **C** for wavelength pair  $\lambda_i$  and  $\lambda_j$  as

$$\mathbf{C}_{i,j} = \sigma_1^2 \exp\left(-\frac{1}{2} \left(\lambda_i - \lambda_j\right)^2 / l^2\right) + \sigma_0^2 \tag{2}$$

where parameter l is a correlation length, parameter  $\sigma_0^2$  is non-spectral (i.e. non-spatial) diagonal variance and  $\sigma_1^2$  is spectral (i.e. spatial) variance. We like to note that our used parameter values are specific for this study with OMI data and have been empirically evaluated. These parameter values were estimated from an ensemble of the residuals, i.e. the differences between the observed and modeled reflectances, as described in the paper Määttä et al. (2014). Here we used l = 90 nm and for  $\sigma_0^2$  and  $\sigma_1^2$  we used values of 1% and 2% of the observed reflectance, respectively.

By Bayes' formula the posterior distribution for  $\tau$  within the model m and given the observed reflectance  $\vec{R}_{obs}$  is

$$p(\tau | \vec{R}_{\rm obs}, m) = \frac{p(\vec{R}_{\rm obs} | \tau, m) \, p(\tau | m)}{p(\vec{R}_{\rm obs} | m)}.$$
(3)

In this case we have one unknown  $\tau$  (i.e. AOD at 500 nm) and the full posterior distribution is calculated as described below.

The posterior is evaluated at a dense grid, e.g. at 200 points, of  $\tau$  values, basically in the range of  $[0, \tau_{\text{max}}]$ . The maximum allowed  $\tau_{\text{max}}$  is determined by the model LUT.

We calculated the likelihood as

$$p(\vec{R}_{\rm obs}|\tau, m) = c \exp(-\frac{1}{2} * \chi^2_{\rm mod}(\tau)),$$
 (4)

where  $\chi^2_{\text{mod}}(\tau)$  is calculated from Eq. 1 for the set of  $\tau$  values in the range of  $[0, \tau_{\text{max}}]$ . The constant *c* ensures that the probability distribution is properly defined and it is the same for all the models *m*.

We assumed that a prior distribution  $p(\tau|m)$  for  $\tau$  within aerosol microphysical model m follows a log-normal distribution

$$p(\tau|m) \propto \log N(\tau_0, \sigma_\tau^2).$$
(5)

This confirms that  $p(\tau|m)$  can take only positive real values and ensures that AOD is positive. We set mean value  $\tau_0 = 2$  for the log-normal distribution.

We calculated the normalizing constant (or scaled factor) of the posterior numerically as

$$p(\vec{R}_{\rm obs}|m) = c \int p(\tau|m) * \exp(-\frac{1}{2} * \chi^2_{\rm mod}(\tau)) d\tau.$$
(6)

Consequently, we have now calculated all the elements of the posterior distribution for  $\tau$  (Eq. 3).

In our study we call  $p(\vec{R}_{obs}|m)$  as the model evidence that is used to make the model selection. We select models with the highest evidence value until the cumulative sum of the selected models' evidences pass the value 0.8 or the number of chosen models is 10.

Next we calculate relative evidence for model  $m_i$  with respect to the other models selected above (max 10) by

$$p(m_i | \vec{R}_{\text{obs}}) = \frac{p(\vec{R}_{\text{obs}} | m_i)}{\sum_i (\vec{R}_{\text{obs}} | m_j)}.$$
(7)

These relative evidence values are used to compare models among the set of selected best fitting models.

The averaged posterior distribution over the selected best models  $m_i$  is calculated as

$$p_{\rm avg}\left(\tau | \vec{R}_{\rm obs}\right) = \sum_{i=1}^{n} p(\tau | \vec{R}_{\rm obs}, m_i) p(m_i | \vec{R}_{\rm obs}), \tag{8}$$

where n is the number of models.

We accept the solution for the pixel if the threshold value  $\chi^2 \leq 2$  calculated by following modified chi-squared formula

$$\chi^2 = \frac{1}{n-1} \vec{R}_{\rm res}(\lambda)^{\rm T} \left( \mathbf{C} + {\rm diag}(\sigma^2(\lambda)) \right)^{-1} \vec{R}_{\rm res}(\lambda).$$
(9)

We do this test only for the best model.

As a summary, we do the following for model selection, calculation of posterior distributions and getting MAP estimate of AOD:

- 1. fit each model from LUT (i.e.  $\vec{R}_{mod}(\tau, \lambda)$ ) in turn to the measured reflectance  $\vec{R}_{obs}(\lambda)$
- 2. for each model, find  $\tau$  that minimizes  $\chi^2_{\rm mod}(\tau)$  (Eq. 1)
- 3. for each model, calculate posterior distribution  $p(\tau | \vec{R}_{obs}, m)$  (Eq. 3)
- 4. use model evidence (Eq. 6) to select max 10 best models
- 5. calculate the relative evidence (Eq. 7) for each model among the selected best models. Actually, we first carry out steps 2.-3. once more for the selected best models and then calculate the relative evidences.

- 6. calculate the averaged posterior distribution (Eq. 8) and get point estimate for AOD, i.e. MAP estimate
- 7. finally, do the goodness-of-fit test (Eq. 9)


$$\boldsymbol{R}_{\text{mod}}\left(\lambda,\tau,\mu,\mu_{0},\Delta\phi,p_{\text{s}}\right) = R_{a}\left(\lambda,\tau,\mu,\mu_{0},\Delta\phi,p_{\text{s}}\right) + \frac{A_{\text{s}}(\lambda)}{1 - A_{\text{s}}(\lambda)s\left(\lambda,\tau,p_{\text{s}}\right)}T\left(\lambda,\tau,\mu,\mu_{0},p_{\text{s}}\right).$$
(1)

In the formula, Here path reflectance Ra, transmittance T and spherical albedo s of the atmosphere are derived by LUT from
LUT by interpolation as a function of λ, τ, Δφ (relative azimuth angle), ps (surface pressure), μ (cosine of viewing zenith angle) and μ0 (cosine of solar zenith angle). The sun-satellite geometry data Δφ, ps, μ and μ0 are included in the OMI Level 1B data. The surface reflectance As is taken from the Lambertian equivalent surface reflectance climatology based on the geolocation of the retrieved pixel and month.

**3.1 Acknowledging the model discrepancy**

15 The aerosol microphysical models used in the retrieval procedure are discrete representations of the aerosols in the real atmosphere. This forward model approximation error, Approximations in forward modeling together with uncertainties in the assumptions, e.g. in the surface reflectance, cause model discrepancy, which manifests itself as systematic deviations between the modeled and observed reflectance.

In general description of a retrieval method, a measurement noise term includes the measurement and forward model error.
 20 We pay special attention to the model discrepancy in the fitting process by adding the related uncertainty term η(λ) to the observation model

$$\boldsymbol{R}_{\text{obs}}(\lambda) = \boldsymbol{R}_{\text{mod}}(\tau, \lambda) + \eta(\lambda) + \epsilon_{\text{obs}}(\lambda).$$
(2)

The new model discrepancy error term  $\eta(\lambda)$  enables correlated errors between neighboring wavelengths, thus allowing for smooth departures from the model. The measurement error term  $\epsilon_{obs}(\lambda) \epsilon_{obs}(\lambda) \sim N(0, \sigma_{obs}^2(\lambda))$  will describe the independent

25 instrument noise that will be assumed to be known in the retrieval procedure coming from the instrument properties and from the calculation of the observed reflectance. In the fitting procedure, for simplicity, we use measurement error  $\epsilon_{obs}(\lambda)$  determined as  $R_{obs}(\lambda)/SNR$ , where SNR = 500 is have  $\sigma_{obs}(\lambda) = R_{obs}(\lambda)/SNR$ , where we used value SNR = 500 for the signal-to-noise ratio of the instrument.

Our approach to estimate the model discrepancy term  $\eta(\lambda)$  was to explore systematic differences between the measured and

30 modeled reflectance (i.e. residuals). The systematic structure in the residuals indicates inadequacy in the forward model. The model discrepancy was characterized using a zero mean Gaussian process  $\eta(\lambda) \sim GP(0, \mathbf{C})$  (Rasmussen and Williams, 2006) $\eta(\lambda) \sim GP(0, \mathbf{C})$ (Rasmussen and Williams, 2006), as described by Määttä et al. (2014). The covariance function, where the covariance matrix C of the Gaussian process defines the defines the wavelength-dependent correlation properties of the discrepancy. We constructed the covariance function empirically by using the wavelength distance dependent correlation structure of the residuals. The covariance matrix C was constructed by means of an empirical semivariogram when the variances of the residual differences were calculated for each wavelength pairs with the distance d. Next, the theoretical Gaussian variogram model was fitted to

5 these empirical semivariogram values. The outcome of this analysis were the values for parameters that defines the model discrepancy covariance matrix C (see Määttä et al. (2014) for details).

As a result, We assume that the likelihood function describing the distribution of the observations given the model follows a Gaussian distribution. The likelihood function has an additional error covariance term due to the model error,

$$p(\boldsymbol{R}_{\text{obs}}|\tau,m) \propto \exp\left(-\frac{1}{2}\boldsymbol{R}_{\text{res}}(\lambda)^T \left(\mathbf{C} + \operatorname{diag}\left(\sigma_{\underset{\sim}{\text{obs}}}^{2}(\lambda)\right)\right)^{-1} \boldsymbol{R}_{\text{res}}(\lambda)\right),\tag{3}$$

10 where  $\mathbf{R}_{res}(\lambda) = \mathbf{R}_{obs}(\lambda) - \mathbf{R}_{mod}(\tau, \lambda)$  is the residual, **C** is the model discrepancy covariance matrix and diag( $\sigma^2(\lambda)$ ) is a diagonal matrix of the measurement error variances  $\sigma^2(\lambda)$ .  $\sigma^2_{abs}(\lambda)$ . The likelihood function is needed for calculation of posterior distribution using Bayes' formula (see Sect. 3.2).

**3.2 Aerosol type and AOD retrieval with uncertainty quantification**

In the Bayesian inference, the solution of an inverse problem is presented as a posterior distribution of the unknown. This approach provides a natural way to present the uncertainty in the AOD and in the aerosol microphysical model m. By Bayes' formula the posterior distribution for  $\tau$  within the model m and given the observed reflectance  $\mathbf{R}_{obs}$  is

$$p(\tau | \boldsymbol{R}_{\text{obs}}, m) = \frac{p(\boldsymbol{R}_{\text{obs}} | \tau, m) p(\tau | m)}{p(\boldsymbol{R}_{\text{obs}} | m)},\tag{4}$$

where  $p(\mathbf{R}_{obs}|\tau, m)$  is the likelihood and  $p(\tau|m)$  is a prior distribution for  $\tau$  depending on the aerosol microphysical model m. The denominator  $p(\mathbf{R}_{obs}|m)$  does not depend on  $\tau$  and acts to normalize the numerator. We assumed that the prior  $p(\tau|m)$

- 20 follows a log-normal distribution in order to ensure that the estimated AOD is positive. The calculation of the actual posterior distribution requires solving integrals over the parameter and model space. In the present In our case, the estimation and model selection procedure seeks the solution for a one-dimensional one parameter  $\tau$ , and the calculations will be and then the calculation of posterior distribution is fairly straightforward by numerical quadrature. The posterior distribution calculation calculation of the posterior distribution is presented in more detail in Määttä et al. (2014).
- 25 The denominator  $p(\mathbf{R}_{obs}|m)$  The denominator  $p(\mathbf{R}_{obs}|m) = \int p(\mathbf{R}_{obs}|\tau, m) p(\tau|m) d\tau$  in Eq. (4) is the probability of the observed reflectance  $\mathbf{R}_{obs}$  assuming the model m is the correct one. However, when considering our problem of choosing the right model m, the  $p(\mathbf{R}_{obs}|m)$  acts as an evidence in favour for m. Consequently, we compare models using their evidence values. In the retrieval procedure we accept the models with the highest evidence until a cumulative sum of the selected models' evidences pass the value of 0.8 or the number of selected models is ten.

Since we assume that a priori all models are equally likely, we end up calculating the relative evidence for each selected model  $m_i$  by formula

$$p(m_i | \mathbf{R}_{\text{obs}}) = \frac{p(\mathbf{R}_{\text{obs}} | m_i)}{\sum_j p(\mathbf{R}_{\text{obs}} | m_j)}.$$
(5)

Here the denominator is a sum over all the evidences of the models  $m_i$  under the comparison process (see Määttä et al. (2014) for details). The relative evidence indicates how plausible the aerosol microphysical model is among the set of potential models.

Even when a model has the highest evidence it does not ensure that it gives an adequate fit to the observed reflectance. The goodness of fit of the selected model is analyzed by the modified chi-squared value

$$\chi^{2} = \frac{1}{n-1} \boldsymbol{R}_{\text{res}}(\lambda)^{\text{T}} \left( \mathbf{C} + \text{diag}(\sigma^{2}(\lambda)) \right)^{-1} \boldsymbol{R}_{\text{res}}(\lambda), \tag{6}$$

where C is a covariance matrix for the model discrepancy and n is the number of wavelength bands in the spectral reflectance. We accepted the retrieved solution (i.e. the selected best model) if this merit function gives a value  $\leq 2$ .

**3.3 **Bayesian model averaging**

5

10

Traditionally, the aerosol microphysical model  $m_i$  with the highest evidence can be treated as the correct one. However, there can be several models that could explain the measurements equally well when taking into account the uncertainty in the selection procedure. In that case the selection of single model (i.e. aerosol sub-type) does not ensure that it is the most

15 appropriate model since it may have been selected by chance. In addition, the posterior distribution for  $\tau$  can differ from model to model among the best models. This indicates that the selection of one particular model as the correct one is not always self-evident or meaningful.

We have used the Bayesian model averaging approach (Hoeting et al., 1999; Robert, 2007) to calculate averaged posterior distribution by formula

20
$$p_{\text{avg}}(\tau | \boldsymbol{R}_{\text{obs}}) = \sum_{i=1}^{n} p(\tau | \boldsymbol{R}_{\text{obs}}, m_i) p(m_i | \boldsymbol{R}_{\text{obs}}),$$
 (7)

where the posterior distributions for  $\tau$ , assuming that  $m_i$  is the correct model, are weighted by the models' evidences. By model averaging we perform the shared inference about the AOD over the best fitting models. Secondly, the uncertainty in the model selection is incorporated in the uncertainty estimate of the AOD.

**4 Case studies and results**

With the following test cases we study functioning of the aerosol type selection procedure, concept of the evidence for model 25 comparison and the resulted AOD posterior distribution for expressing the uncertainty due to model selection and approximations in forward modeling. The relative evidence of a single model, with respect to the other selected models, describes the plausibility of that model to explain the observed reflectance. The width of the posterior density function illustrates the level of the uncertainty, i.e. the wider the width the higher the uncertainty.

We consider two test cases where the atmospheric aerosol conditions are different from each other. The first case study focuses on an urban area around Beijing, where we analyze the retrieved aerosol characteristics on two days to observe the difference as well as similarity of aerosol conditions in these days. The Beijing case is challenging since the aerosol air-mass is a mixture of dust from north blending with urban pollution around Beijing. On the other hand, this case enables to examine

5 aerosol type selection in situation with high AOD levels. The other test case cover covers Northern and Central Africa where we expect dust aerosols in the north and biomass burning aerosols in the central part. In particular, this test case cover covers a large, almost cloud free, area.

We evaluated the retrieved AOD estimates using collocated ground-based Aerosol Robotic Network (AERONET) data of aerosol properties. The AERONET program is a federation of ground-based remote sensing aerosol networks (Holben et al.,

10 1998). We downloaded the Version 2 Direct sun Level 2.0 quality assured and cloud-screened aerosol data for the AERONET sites under investigation. We tracked clouds and land scene for the case studies by utilizing true-color images from MODIS, on Aqua satellite, that has the equator crossing time only about 15 minutes earlier than OMI. The MODIS instrument is onboard both Terra and Aqua spacecraft. The data products derived from MODIS measurements include atmosphere (e.g. cloud mask and aerosol products), land, cryosphere and ocean products (see e.g. http://modis.gsfc.nasa.gov).

**15 4.1 Beijing area on 16 April and 27 April 2008**

In this case study focusing on an urban area around Beijing, we analyzed the retrieved aerosol characteristics on two days: the 16th and 27th of April 2008. In the spring season the atmosphere is typically loaded by a mixture of urban and dust aerosols (Yu et al., 2016). Figure 1 shows the true-colour images from the MODIS, on-board the NASA's Aqua satellite, on the 16th of April 2008 at 05:15 UTC (left) and 27th of April 2008 at 04:55 UTC (right) over the Beijing area.

The OMI pixels that were analysed are located on rows 23-29 across the orbit in the first day case (i.e. the 16th of April) and on rows 10-20 in the second day case (i.e. the 27th of April), respectively. The pixel has no data if it No data are reported if the pixel is cloud contaminated or none of the models had adequate fit with the measured reflectance (Eq. 6).

Figure 2 presents the number of most appropriate models retrieved for each pixel on both days. The maximum number of best models was restricted to ten (see Sect. 3.2). In the first day case the variety of the number of best models is wide (left)

25 whereas in the latter day case (right) for the most part of the pixels the maximum number of models are selected to explain the measurements.

In Fig. 3 is shown the distribution of the main aerosol types of the retrieved aerosol microphysical models (i.e. sub-types) having the highest evidence. The main aerosol types are the weakly absorbing (WA), weakly absorbing sea salt (WA1114), biomass burning (BB), desert dust (DD) and volcanic (VO) aerosols. The prevailing types in both days in the vicinity of Beijing

30 AERONET site (marked with black star in Fig. 3) are the BB and WA types. The appearance of marine type 'WA1114' as the best matching type may occur due to cloud impact since the nearby pixels with no results have been omitted as cloudy pixels. In addition, on the 27th of April (right) the desert dust type gets the highest evidence in the upper part of the examination area. An airmass trajectory analysis (not shown here) indicated that on the 27th dust from the Gobi desert (north of the study area) was entering the Beijing area.

Figures 4 and 5 illustrate for both days how plausible each main aerosol type is to represent the prevailing aerosol air-mass type. We have summed up the relative evidences (%) of the selected models (i.e. sub-types) within each main type to get a quantity of confidence, i.e. shared evidence, for each main aerosol type. Figures b-f show pixel wise pixel-wise the shared evidence (%) for each main type. Whereas, the figure on the upper left corner (a) presents the relative evidence (%) of the

- 5 single best fitting aerosol microphysical model indicating how superior the ranked best model is with respect to the other selected models if any. We can notice that the one best model does not necessarily determine the aerosol type alone, but a mixture of models could give a better match. We also observe that both WA and BB type aerosol microphysical models get support as representative models for some pixel areas (b-d). In addition, in the latter day case on the 27th, the DD type gets strong evidence in the upper right corner of the examination area (Fig. 5e).
- 10 Figure 6 shows the distribution of the retrieved MAP AOD estimates in both days. The upper row show the MAP estimate from the aerosol microphysical model with the highest evidence. The lower row show the MAP estimate from the averaged posterior distribution over the selected best models. In general, the AOD point estimate value from the averaged posterior distribution is lower than the AOD estimate based on the single best model.
- Figure 7 shows the results for a single pixel having a geometric collocation with the AERONET Beijing site, i.e. the Beijing site coordinates are inside an OMI pixel. The upper row shows the measured reflectance (in blue) and the selected, best matching, modeled reflectances (in green) for the 16th of April (left) and the 27th of April (right). In the lower row are shown the posterior density functions that characterize the uncertainty. Also, the best matching models' identification numbers and the associated relative evidences inside brackets are given. The relative evidence (%) (Eq. 5) express how plausible this model is to explain the measured spectral reflectance with respect to the other selected best models. The averaged posterior distribution
- 20 (Eq. 7) has two peaks indicating difficulty in model selection. The red vertical dashed line denotes the MAP AOD estimate (i.e. the posterior mode) from the averaged posterior. The grey vertical lines show the AERONET AOD at 500 nm values at separate measurement times.

On the 16th of April (Fig. 7 left) there are two best matching models, both of the BB type, selected. The other types of models, e.g. weakly absorbing type, do not match as well as the selected best BB models. The width of the averaged posterior

25 is relatively wide indicating high uncertainty in the result. The model with the higher evidence has much weight in the averaged posterior and this affects the retrieved AOD that is higher than the AERONET values. The AERONET measurements are in the time range of 00:02-04:59 UTC therefore before the OMI overflight time (~ 5:25 UTC). However, there are some AERONET AOD measurements (n=3) within two hours time window including OMI overpass time. These AOD values are marked by darker grey vertical lines and the black vertical line is the average. Figure ?? shows the best fitted modeled reflectances when

30 we considered only weakly absorbing type models' LUTs. We can see that the match with the observed reflectance is not as good as the selected best BB models have (Fig. 7 left).

On the 27th of April (Fig. 7 right) we can be confident with the result as it the resulted AOD from the averaged posterior is in agreement with the AERONET data. There is one WA type model (blue posterior curve) ranked as the second best model in the fitting. At that time the AERONET measurement In that day the AERONET measurements, marked by grey vertical lines, are

in time range of 08:22-09:43 UTC, thus monitored after the OMI overpass time ( $\sim$  5:06 UTC). The selected best models are

BB type, except one WA type model (blue posterior curve) that is ranked as the second best model in the fitting. The averaged posterior distribution has two modes indicating two alternative explanations for the observed reflectance. The higher mode of the averaged posterior gets a larger portion of the model evidences thus yielding the MAP AOD estimate. But, if the end result is based on the one best fitting model, i.e. "BB2312", the estimated AOD level would be higher since the corresponding

5 posterior distribution curve is the most right one.

In Table 2 are given the aerosol characteristics for the AERONET sites and results, e.g. AOD at 500 nm and Ångström exponent values, retrieved by the proposed method. The AERONET data for Beijing shown in Table 2 are the daily averages. We interpolated AOD at 500 nm by using the AERONET AOD at 440 nm and the AERONET provided Ångström exponent 440-675 nm. The Ångström exponent describes the dependency of the AOD on wavelength. It gives an approximation of the

- 10 aerosol particle size in such a way that when coarse aerosol particles dominate the exponent is small, and vice versa for the fine particle dominance. In our retrieval we calculated the Ångström exponent (442-500 nm) by Ångström exponent power law where the AOD at 442 nm was derived from the associated LUT based on the retrieved AOD at 500 nm. Thus the reported Ångström exponent is completely determined by the model LUT. In Table 2 are presented Ångström exponent values based on the best fitted ( $\alpha$ 1) and the second best fitted ( $\alpha$ 2) model. For Beijing case, in both days, the derived Ångström exponent value
- 15 of the best model ( $\alpha$ 1) is in good agreement with the AERONET value. Even so, on the 16th of April  $\alpha$ 2 deviates more from the AERONET value although the estimated AOD, based on the second best model, is closer to the AERONET AOD values (see Fig. 7 left).

**4.2 Northern and Central Africa on 26 March 2008**

This case study covers a large area over the Northern and Central Africa on the 26th of March 2008. Figure 8 shows the MODIS
true-colour images on the 26th of March 2008 at 13:00 and at 13:05 UTC over the Northern and Central Africa. The view is mainly cloud free except for some broken-cloud cover in coastal regions. The AOD data from four AERONET sites Agoufou (North Mali), DMN\_Maine\_Soroa (Niger), IER\_Cinzana (Mali) and Saada (Morocco) are used to evaluate the results. Daily averaged AERONET Version 2 Direct sun Level 2.0 AOD data are reported in Table 2.

In Fig. 9 we can see the areas where the maximum number of aerosol microphysical models are selected, as well as the areas where only one model dominates. In the middle of the orbit is an area where none of the models has adequate fit with the measured reflectance (Eq. 6). In March 2008 the rows 54-55 (i.e. 53-54 if 0-based) in the OMI measurements are affected by a row anomaly (OMI row anomaly team, 2016). We have omitted these two rows in the analysis.

As seen in Fig. 10 the desert dust is the dominating type of the selected best models. There are also areas where the BB type of models get the highest evidence to explain the measurement. The pixels where the weakly absorbing sea salt type aerosols

30 (type WA1114) have the best fit are located in the edge areas of clouds or in partly cloudy areas (see Fig. 8). The unexpected selection of volcanic aerosol type appears as the only best aerosol type in the appropriate aerosol type happens for pixels located northeast from the Lake Chad . The selection of volcanic type is most probably related to a white area seen where is seen cloud in the MODIS RGB image (Fig. 8(left)...). The location of AERONET sites are marked with black stars.

The Figs. 11b-f reveal that all the selected best models (i.e. sub-types) usually are of the same main type, namely BB or DD type. But, in the area around Algeria occurrence of a mixture of main types may be related to the cloud contamination of these pixels.

From the Fig. 12 it can be concluded that often the MAP estimate from the single best aerosol microphysical model has slightly higher AOD level (left) than the MAP estimate from the averaged posterior distribution over the selected best models

- (right). The retrieved AOD estimates are rather small perhaps indicating that no dust event or active fires were going on.
  However, we We can notice that in the area south of the latitude of 8°, where the biomass burning type of models dominate, the AOD estimates are higher. Active fire maps from satellite data (not shown here) support the fire activity at that area. We can also notice the DD type dominating area near the Agoufou AERONET site where AOD level is higher (see Fig. 12 and
- 10 Fig. 11e).

5

Figure 13 shows the Ångström exponent (442-500 nm) values based on the best fitted (left) and the second best fitted (right) model. In the Table 2 are shown the calculated Ångström exponent (442-500 nm) values at the locations of the reference AERONET sites. Again, it should be noted here that the retrieved Ångström exponent is completely determined by the model LUT. Consequently, as seen in Fig. 13 the Ångström parameters reflect the selected aerosol microphysical models. That is, the

15 Ångström values are low where the desert dust type of models dominate. Correspondingly, in the coastal region where typically is smoke and urban polluted air the Ångström exponent is higher.

Figures 14 and 15 show the spectral reflectance fitting curves (on the left hand side) and the retrieved AOD estimates with uncertainty (on the right hand side) for the single pixels located around the AERONET sites: Agoufou, DMN\_Maine\_Soroa, IER\_Cinzana and Saada. In the figure showing the posterior distributions (right) the grey vertical lines indicate AERONET

20 Direct sun Level 2.0 AOD at 500 nm values measured during that day. The darker grey vertical lines denote AERONET AOD values within two hour time window including the OMI overpass time and the black vertical line is the average of these AOD values.

The measured reflectance at the Agoufou site (first row in Fig. 14) has a rather unique spectral structure but there is and there is only one dust type model that fits to the measured reflectance adequately well. at all. The associated AOD from the

- 25 model LUT is unreasonable high with respect to the AERONET values. Also the The large width of posterior the posterior distribution, that is the averaged posterior distribution as well, indicates high uncertainty --in the model selection and thus in the retrieved AOD. Even the retrieval uncertainty is high, the posterior density does not cover the AERONET AOD values at all. But the posterior density covers AOD from the OMAERO product that is also high compared to AERONET daily average value (Table 2). It can be noted here that AERONET measurements at the Agoufou site were made in the morning in time range
- 30 of 06:58-09:27 UTC whereas the OMI overpass time at that location was at ~ 13:15 UTC. However, the derived Ångström exponent  $\alpha$ 1 has rather good agreement with the AERONET value (Table 2).

At the other three reference AERONET sites there are measured AOD values during the OMI overpass time. In Fig. 14, showing the results for DMN\_Maine\_Soroa case (lower row), all the ten selected models belong to the biomass burning type and their posteriors indicate an uniform small uncertainty. The estimated AOD values are consistent with the AERONET AOD

values within uncertainty but the derived Ångström exponents,  $\alpha 1$  and  $\alpha 2$ , do not agree with the AERONET value (Table 2). In

the IER\_Cinzana case (Fig. 15 upper row) the selected models are the desert dust type (orange) except for one biomass burning type model (red curve). The estimated AOD is a bit lower than the AERONET measurements. In the Saada case (Fig. 15 bottom row) the AERONET AOD values are in good agreement with estimated AOD, although the averaged AERONET AOD (black vertical line) of measurements the measurements made around the OMI overpass time is slightly lower than the MAP

5 estimate of AOD (red dashed vertical line). Also, the AOD values from OMAERO product are in good agreement with the AERONET values (Table 2). For the sites IER\_Cinzana and Saada the best and the second best models have as good evidence (Fig. 15 right column) indicating that the selection of the best model happened by chance. Consequently, the derived  $\alpha$ 1 for Saada site is consistent with the AERONET value whereas the derived  $\alpha$ 2 for IER\_Cinzana has better agreement than  $\alpha$ 1 with the AERONET value.

**10 5 Discussion and Conclusions**

In this paper, we focused on the aerosol microphysical model selection in the aerosol retrieval and on the quantification of uncertainty for the retrieved aerosol type and AOD using OMI TOA spectral reflectance measurements. The retrieval aerosol type selection from LUTs is a source of uncertainty and affects the accuracy of the retrieval. The main targets of our study are 1) to improve the retrieval error estimate (i.e. to produce more realistic uncertainty estimate), 2) to evaluate the model

- 15 choice procedure and 3) to find more robust AOD estimate that is based on the average of the most appropriate aerosol microphysical models instead of on a single model chosen probably by chance. The retrieval scheme is similar to the OMAERO algorithm using information from several wavelength bands between 330 and 500 nm and pre-calculated LUTs for aerosol microphysical properties. The presented methodology was introduced and previously used by Määttä et al. (2014) for the uncertainty quantification in the retrieval of AOD at the reference wavelength of 500 nm. This new research investigates the
- 20 uncertainty in the aerosol type selection in more detail. What's more, we experimented the proposed methodology with test cases covering large pixel areas. We evaluated the retrieved AOD by comparison with AERONET measurements at example sites. For simplicity, we studied only cloud-free over-land OMI pixels.

The method uses Bayesian statistical inference to quantify uncertainties due to model selection and due to approximations in the forward modeling. The concept of model evidence is used as a tool for model comparison and to assist in the selection of the best models. The forward model approximations cause model error that results in systematic differences between the modeled and observed reflectance. We acknowledge this model discrepancy when choosing the most appropriate LUTs in order to produce more realistic uncertainty estimates of the retrieved AOD. Following the Bayesian approach the uncertainty is described by the posterior probability distribution. The selection of single best fitting aerosol microphysical model is not always clear and this uncertainty is addressed in this study. We use a statistical technique based on Bayesian model averaging

30 to combine the AOD posterior probability densities of the best models to obtain the averaged posterior distribution. Then the retrieved AOD is the MAP estimate of the averaged posterior function. We also determine the shared evidence of the best matching models within a main aerosol type (weakly absorbing, sea salt, biomass burning, dust and volcanic) in order to quantify plausibility of each main aerosol type.

Retrieving the aerosol type and AOD from the TOA reflectance measurements is an ill-posed problem and a priori information of prevailing aerosol conditions are needed to get a solution. The limited information content in the OMI measurements and the narrow wavelength band range ending up to 500 nm make the problem very challenging to solve. We investigated the developed method by studies covering several larger pixel set areas. We evaluated the retrieved AOD by comparison with

**5 AERONET measurements at example sites.**

In our approach we did not pre-select aerosol microphysical models based on e.g. a climatology of aerosol geographical distribution. Instead, we fitted all the available models (i.e. LUTs), a total of fifty models, to the spectral measurement. This makes the whole process slower but is justified here to study the uncertainty in the aerosol model selection. The goodness-of-fit value (Eq. 6) was used to analyze whether the retrieved solution is acceptable. We used a limited set of aerosol microphysical

- 10 models, a total of fifty models. It is highly likely that the used model set is not comprehensive enough to represent all aerosol air-mass conditions. In the Beijing case studies the absorbing biomass burning type aerosol microphysical models dominate (Sect. 4.1) and the reason could be the lack of proper models for the prevailing aerosol conditions during the selected days. In addition, in the Northern and Central Africa case (Sect. 4.2) there is an area in the middle of the orbit where none of the available models gave an adequate fit.
- We made the cloud screening in a straightforward way just using the effective cloud fraction threshold value of 0.34 (see Sect. 2) and thus there were most probably cloud affected pixels left. The suspicious results were often localized to pixels in the edge of a cloud or inside broken-cloud areas. In these cases the observed reflectance was such that the unexpected model, e.g. an oceanic or volcanic aerosol model, had the best fit -(see Fig. 10). This feature could be used as an additional cloud detection.
- Here the model discrepancy was determined empirically by exploring a set of residuals (i.e. the difference between observed and modeled reflectance) and then fitting a Gaussian process to find the characteristics of the model error (see Määttä et al., 2014). Brynjarsdóttir et al. (2014) discussed model discrepancy and its effect on results with simple examples. They also emphasized the importance of modeling the model error properly and the use of realistic priors for the model discrepancy.

The aerosol type selection from LUTs is a source of uncertainty and affects the accuracy of the retrieval. The aim of our

- 25 study was to produce more realistic uncertainty estimates. As a result, we can account for The presented method accounts for the forward modeling uncertainty and also includes the uncertainty due to model selection in the total uncertainty budget. In particular, it gives tools for analyzing the different sources of uncertainties and the model error and also include the model selection uncertainty in the total uncertainty budgetinfluence of aerosol microphysical model selection on the estimated AOD. The case studies indicate that the developed methodology, in general, works in the varying aerosol conditions as expected. But,
- 30 even the method gives a solution that passed the goodness-of-fit test it does not ensure correctness of the result. We found that the increased uncertainty of AOD expressed by the posterior distribution reflects the difficulty in model selection. The posterior probability distribution can characterize the uncertainty more extensively than commonly given standard deviation. This brings more information about the uncertainty and produce more realistic uncertainty estimate, as well. We can also conclude that, for the most part, the combination of aerosol models obtained by the Bayesian model averaging approach gives better AOD
- 35 estimate than if based on one best model that may have been selected by chance. There are cases when several selected models

have almost the same portion of the relative evidence and then the order of the best models may have been happened by chance. This situation is even more complicated if the level of the AOD varies from model to model.

The comparison with the AERONET data revealed that if the estimated AOD at 500 nm from the averaged posterior distribution is not consistent with the AERONET AOD values the derived uncertainty of AOD is higher. Then, in the most

5 cases, the averaged posterior density covers the AERONET AOD values. The derived, even if LUT dependent, Ångström exponent values are in rather good agreement with the AERONET values (see Table 2). In general, the retrieved AOD values are also consistent with the AOD from the OMAERO product and the solution is achieved for a larger pixel set.

However, in order to confirm robustness of the methodology more comprehensive and systematic assessment and evaluation of the method are needed. This involves validation studies with reference data as well as implementation of the method using

10 other instruments' measurements. Also, the examination of the method with simulated data would bring additional information about usability and reliability of the approach. Moreover, further study and discussion is needed to determine how to express the uncertainty information, provided by the posterior distribution, in more compact form.

The method described in this work is applicable to any instrument measurements where the observed reflectance is available as well as the aerosol microphysical models. Our plan is to apply this method to an AATSR retrieval algorithm where the models

15 are constructed during the fitting using a limited amount of aerosol components describing non-absorbing and absorbing fine particles together with coarse marine and dust particles (Kolmonen et al., 2016).

Acknowledgements. This work was supported by the Academy of Finland INQUIRE project and by the European Space Agency as part of the Aerosol\_cci project. The OMI data was\_were extracted via the NASA's GES DISC Mirador data access system. The authors like to thank the OMI Science Team for providing the data. The authors would like to give many thanks to the KNMI OMI team for providing the

20 aerosol model look-up tables (LUTs) used in this study and discussions on the OMAERO algorithm. The authors also thank the Principal Investigators and their staff for establishing and maintaining Beijing, Agoufou, DMN\_Maine\_Soroa, IER\_Cinzana and Saada AERONET sites. The authors give special thanks to Anu-Maija Sundström (FMI) for the help with figures. The authors are grateful to the two anonymous reviewers for the useful comments to improve the manuscript.

20

Beijing, 16 April. The modeled reflectances from weakly absorbing and marine WA1114 type models' LUTs are fitted to the observed reflectance.